solar system/geochemistry

zircon, lunar samples, Apollo 16, regolith breccias, Pb-Pb age dating, shock-zircon

**Author for correspondence:**
K. H. Joy
e-mail: katherine.joy@manchester.ac.uk

One contribution to the Astronomy and Astrophysics New Talent collection.

# Timing of geological events in the lunar highlands recorded in shocked zircon-bearing clasts from Apollo 16

K. H. Joy[1], J. F. Snape[2], A. A. Nemchin[2,3], R. Tartèse[1], D. M. Martin[4], M. J. Whitehouse[2], V. Vishnyakov[5], J. F. Pernet-Fisher[1] and D. A. Kring[6]

[1]Department of Earth and Environmental Sciences, School of Natural Sciences, The University of Manchester, Manchester, UK
[2]Department of Geosciences, Swedish Museum of Natural History, Stockholm, Sweden
[3]Department of Applied Geology, Curtin University, Perth, Australia
[4]European Centre for Satellite Applications and Telecommunications (ECSAT), European Space Agency, Fermi Avenue, Harwell Campus, Didcot, Oxfordshire OX11 0FD, UK
[5]School of Computing and Engineering, University of Huddersfield, Huddersfield, UK
[6]Center for Lunar Science and Exploration, Lunar and Planetary Institute, Universities Space Research Association, Houston, TX 77058, USA

KHJ, 0000-0003-4992-8750; JFS, 0000-0002-1804-420X

Apollo 16 soil-like regolith breccia 65745,7 contains two zircon-bearing clasts. One of these clasts is a thermally annealed silica-rich rock, which mineralogically has affinities with the High Alkali Suite (Clast 1), and yields zircon dates ranging from 4.08 to 3.38 Ga. The other clast is a KREEP-rich impact melt breccia (Clast 2) and yields zircon dates ranging from 3.97 to 3.91 Ga. The crystalline cores of both grains, which yield dates of *ca* 3.9 Ga, have undergone shock pressure modification at less than 20 GPa. We interpret that the U-Pb chronometer in these zircon grains has been partially reset by the Imbrium basin-forming event when the clasts were incorporated into the Cayley Plains ejecta blanket deposit. The zircon grains in Clast 1 have been partially decomposed, resulting in a breakdown polymineralic texture, with elevated U, Pb and Th abundances compared with those in the crystalline zircon. These decomposed areas exhibit younger dates around 3.4 Ga, suggesting a secondary high-pressure, high-temperature event, probably caused by an impact in the local Apollo 16 highlands area.

# 1. Introduction

The ancient (greater than 4.35 Ga) lunar feldspathic crust has been extensively modified by impact cratering and basin formation, as

**Figure 1.** Lunar nearside context of the Apollo 16 landing site (A16) in comparison to the other sample return sites (A = Apollo, L = Luna) and major impact basins. Clementine albedo image (stereographic projection) is overlain by the concentration of Th (ppm) as mapped by the Lunar Prospector γ-ray mass spectrometer ([41]; 2° per pixel calibration). Only Th concentrations greater than 2 ppm are displayed, showing the extent of the Procellarum KREEP Terrane and KREEP-rich Imbrium basin proximal ejecta blanket.

well as by extrusive and intrusive magmatic activity (e.g. [1] and references therein). Temporal relationships between these endogenic and exogenic processes are preserved in the isotopic record of mineral grains and rock fragments collected from the lunar surface regolith by the Apollo astronauts and robotic Luna sample return missions. Unravelling this polyphased geological record is challenged by several issues. For example, there is a paucity of datable mineral phases that can be targeted multiple times by *in situ* analytical techniques (e.g. [2]) and the small mass and fine-grained nature of the rocks recovered often limits the possibility for repeat analysis of multiple isotope systems, or inter-laboratory measurements of the same bulk sample [3]. There are also difficulties in disentangling magmatic chronological information from isotopic reset events impinged by elevated impact shock pressure and/or secondary metamorphic heating processes [3–8]. More widely there is a lack of clear understanding of the relationship of rock fragments collected in the regolith to their parent bedrock lithology or impact crater formation setting(s) (e.g. [9–11]). Thus, although the knowledge we have gained through sample studies about the Moon's crust formation, impact bombardment and magmatic history is extensive (e.g. [12]), there are still many outstanding questions about the contemporaneous age relationship between primary ferroan anorthosite (FAN) crustal rocks and intrusive rocks of the Mg-Suite and High Alkali Suite (HAS), and the timing and duration of impact basin formation (e.g. [1,3,13–23]).

Ancient crustal and impact rocks were sampled by the Apollo 16 mission to the Cayley Plains and Descartes Mountains in the nearside central region of the Moon (8.9734° S, 15.5011° E; [24]). The landing site sits on the distal ejecta blanket of the Imbrium basin, an event that extensively reworked and mixed in underlying crater, basin ejecta and megaregolith units [25–40]. Geochemically, the proximal and distal Imbrium ejecta blanket can be traced remotely using the abundance of the element Th (figure 1), which is a proxy for the signature derived from instantaneous impact flash melting of KREEP-rich lithologies (e.g. HAS, Mg-Suite and KREEP basalts) under the Imbrium impact structure. The Apollo 16 landing site sits on the periphery of this Th anomaly (figure 1); collected samples include a range of incompatible trace element (ITE)-rich mafic impact melts [31] and rarer KREEP-rich rock fragments (e.g. granitic glasses: [42]; alkali gabbronorites: [43,44]). Many of these

samples are fine-grained and glassy, as such very few ITE-rich Apollo 16 mineral phases have ever been reported or dated [45–49].

Here, we report the first U-Pb dates obtained on three zircon grains found within an Apollo 16 regolith breccia sample. Previously, zircon and other Zr-rich mineral phases like baddeleyite and zirconolite grains have been dated in rock and soil samples from Apollo 12, 14, 15, 17 and in KREEP-rich lunar meteorites (e.g. [46,50–64]). These minerals typically preserve U-Pb dates between 4.4 and 3.8 Ga, associated with either ancient KREEP-driven magmatic episodes or formation/reset during large, high-temperature, impact cratering/basin-forming events (i.e. [10] and references therein; [65]). Zircon and other Zr-rich mineral phases, thus, have the potential to probe early lunar chronological events and shed light on geological processes in the central nearside highlands of the Moon [66].

# 2. Methods

A 30 µm thick polished section 65745,7 was allocated by CAPTEM (Curation and Analysis Planning Team for Extraterrestrial Materials). The section had previously been studied by Simon *et al.* [67]. The whole section was carbon coated and X-ray maps of the whole sample were collected using the NASA Johnson Space Center (JSC) JEOL 6340f field emission gun scanning electron microscope (FEG-SEM) using a beam current of 30 nA and an accelerating voltage of 15 kV to derive 1 micrometre per pixel spatially resolved element distribution maps (figure 2). This initial characterization was conducted as part of locating meteorite fragments as reported in Joy *et al.* [69], and resulted in the discovery of three large zircon grains. Additional back-scattered electron (BSE) images and X-ray maps (collected using an EDAX electron dispersive spectroscopy (EDS) system) of these phases were collected using the NASA JSC JEOL 5910LV SEM (figures 3–5) and a Phillips FEI XL30 ESEM-FEG at The University of Manchester (figure 6). Cathodoluminescence (CL) images and further BSE images (figures 3 and 4) were collected at Manchester Metropolitan University using the Zeiss Supra VP40 FEG-SEM and Gatan MonoCl3+ instruments of the Dalton Research Institute Analytical Microscopy (DRIAM) system.

Silicate mineral and glass chemical composition (electronic supplementary material, tables S1 and S2) were determined using the NASA JSC and The University of Manchester CAMECA SX100 electron probe microanalyser (EPMA) instruments using a 1 or 20 µm beam size, a beam current of 20 nA and an accelerating voltage of 15 kV. Counting times on element peaks were 10 s for Na, 30 s for Mn and Co and 20 s for other elements. Zircon phase (table 1) chemical composition was measured at The University of Manchester using a CAMECA SX100 EPMA using a 1 µm beam diameter, a beam current of 20 nA and an accelerating voltage of 15 kV. REE-doped glasses were used as standards, and analysed on LLIF (Ce, Nd, Eu, Sm, La) and LIF (Lu, Yb) crystals. Other elements were run on LIF (Fe), LPET (Sc, Cl, Ba, Th, Pb, U), PET (K, Ti, Ca, Y and Zr) and TAP (Si, Mg, P, Al, Hf) crystals (see electronic supplementary material, note 1 for full standard set-up). All elements were analysed with 30 s counting time on peaks, apart from K, Ti, Ca, Mg, P and Al, which were counted for 20 s, and Cl that was for 10 s. Elements below detection limits (including Cl, Pb, U and all the REE data) or not actually run during this analytical session for zircon analyses (P, Ba) were discounted from the reported zircon analyses. Well-characterized natural mineral, doped glass and pure metal standards were used to calibrate the other phase EPMA measurements. Data are presented in summary in table 1, and fully in electronic supplementary material, tables S1–S3.

Fourier transform infrared (FTIR) spectroscopy was carried out at The University of Manchester using a PerkinElmer Spotlight400 FTIR spectrometer. Data were collected on non-carbon-coated surfaces using non-polarized reflectance mode. The FTIR instrument was calibrated with a polished gold-coated aluminium reflectance standard, and the environmental background measurement (taken using the gold-coated standard) was automatically subtracted from the spectra prior to co-addition (see [70] for more details). We collected data between 4000 and 650 cm$^{-1}$ wavenumber (approx. 3–15 µm wavelength) with a spectral resolution of 4 cm$^{-1}$. Each pixel was integrated over 64 repeated scans using 'Image-mode' where an array of 16 small (25 µm) detectors were used to spatially resolve pixels with dimensions of 6.25 × 6.25 µm. Data are presented in electronic supplementary material, table S4.

The thin section was then gold coated with approximately 30 nm of gold for secondary ion mass spectrometry (SIMS) analyses. Analyses of the U-Th-Pb systematics in the 65745 zircon grains were first performed using a CAMECA IMS 1280 ion microprobe at the NordSIMS facility in the Swedish Museum of Natural History. The SIMS methodology closely followed the description published elsewhere (e.g. [71,72]). Oxygen was introduced into the sample chamber to enhance Pb$^+$ yield. The mass filtered $^{16}O_2^-$ primary ion beam, with an intensity of 0.5 nA, was reduced through a 50 µm

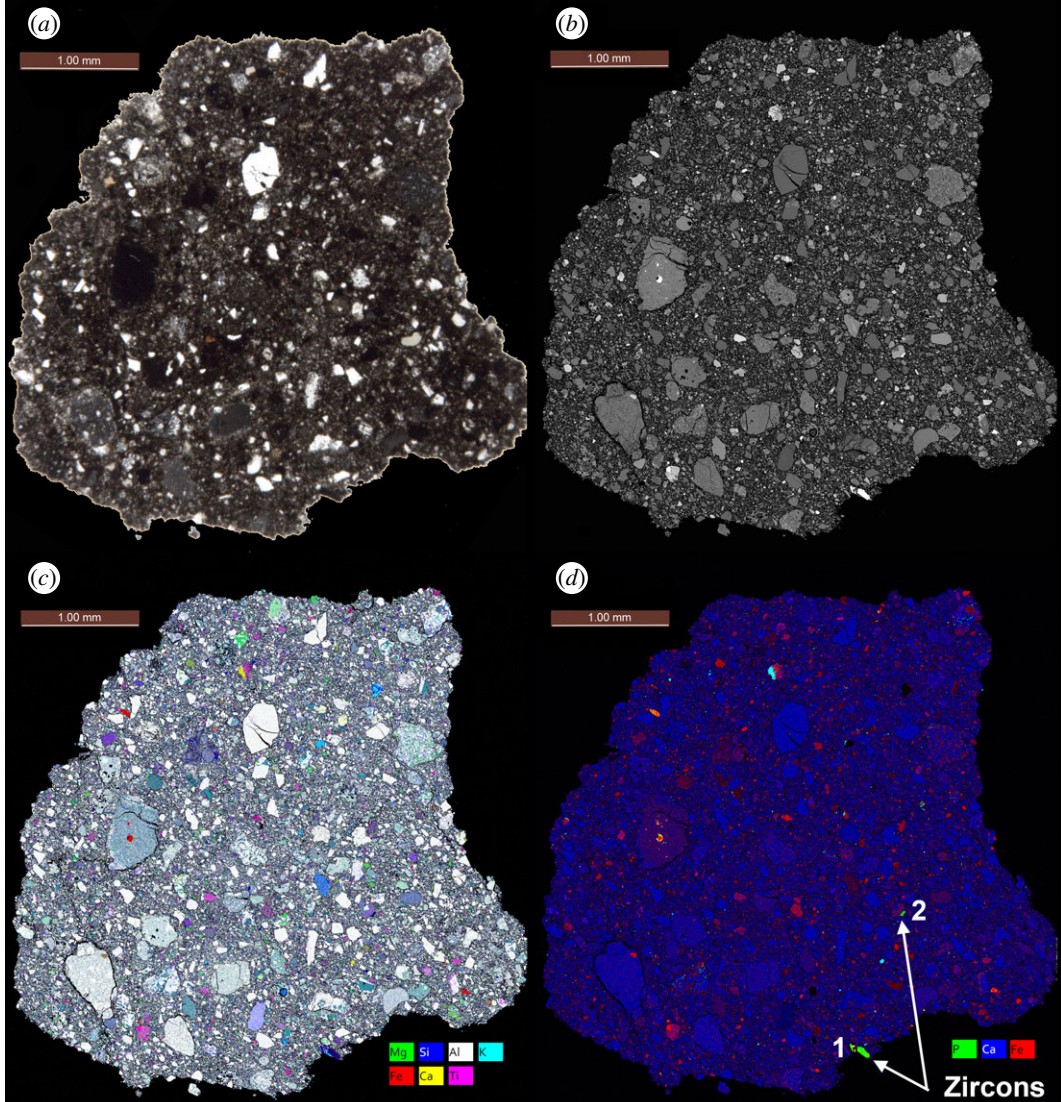

**Figure 2.** Apollo 16 regolith breccia 65745,7. (*a*) Optical image scan of the section surface. (*b*) Montaged area back-scattered electron (BSE) map. (*c*) False colour X-ray element map where Al = white, Ca = yellow, Fe = red, Si = blue, Mg = green, Ti = pink and K = cyan. (*d*) False colour X-ray element map where Ca = blue, Fe = red and P = green. In this colour scheme, phosphates appear cyan, zircon is green and sulfides are yellow (after [68]). The location of Clasts 1 and 2 are denoted.

Köhler aperture to obtain a spot size of 5 µm. An area of 10 µm was presputtered for 70 s before each analysis to remove the gold coating from the area around the analytical spot, and to limit the effects of surface contamination. This presputter was followed by automatic centring of the secondary ion beam in the 4000 µm field aperture and automatic centring of the magnetic field. The mass spectrometer was operated with a nominal mass resolution of 5400 (M/ΔM), sufficient to resolve lead from known molecular interferences. Secondary ion beam intensities were measured with a single low-noise ion-counting electron multiplier in a mass-switching sequence that included $^{90}Zr_2^{16}O^+$ (the matrix peak), $^{204}Pb^+$, $^{206}Pb^+$, $^{207}Pb^+$, $^{208}Pb^+$, $^{232}Th^+$, $^{238}U^+$, $^{232}Th^{16}O^+$ and $^{238}U^{16}O^+$. The U/Pb ratios in zircon were corrected against the 564 Ma zircon CZ3 [73]. The data were corrected for the effects of contamination from terrestrial common Pb using the model values of Stacey & Kramers [74] for present-day terrestrial Pb. Unless stated otherwise, the U-Pb dates have been reported in the following discussion with their associated 2σ uncertainties.

We then mapped the distribution of selected elements at the boundary between the decomposed area and crystalline zircon in Clast 1 Grain 1 using the CAMECA NanoSIMS 50 L instrument at The University of Manchester. Analyses were carried out over a 20 × 20 µm area, divided in 256 × 256 pixels, using an O⁻ primary beam current of approximately 0.5 pA with an accelerating voltage of

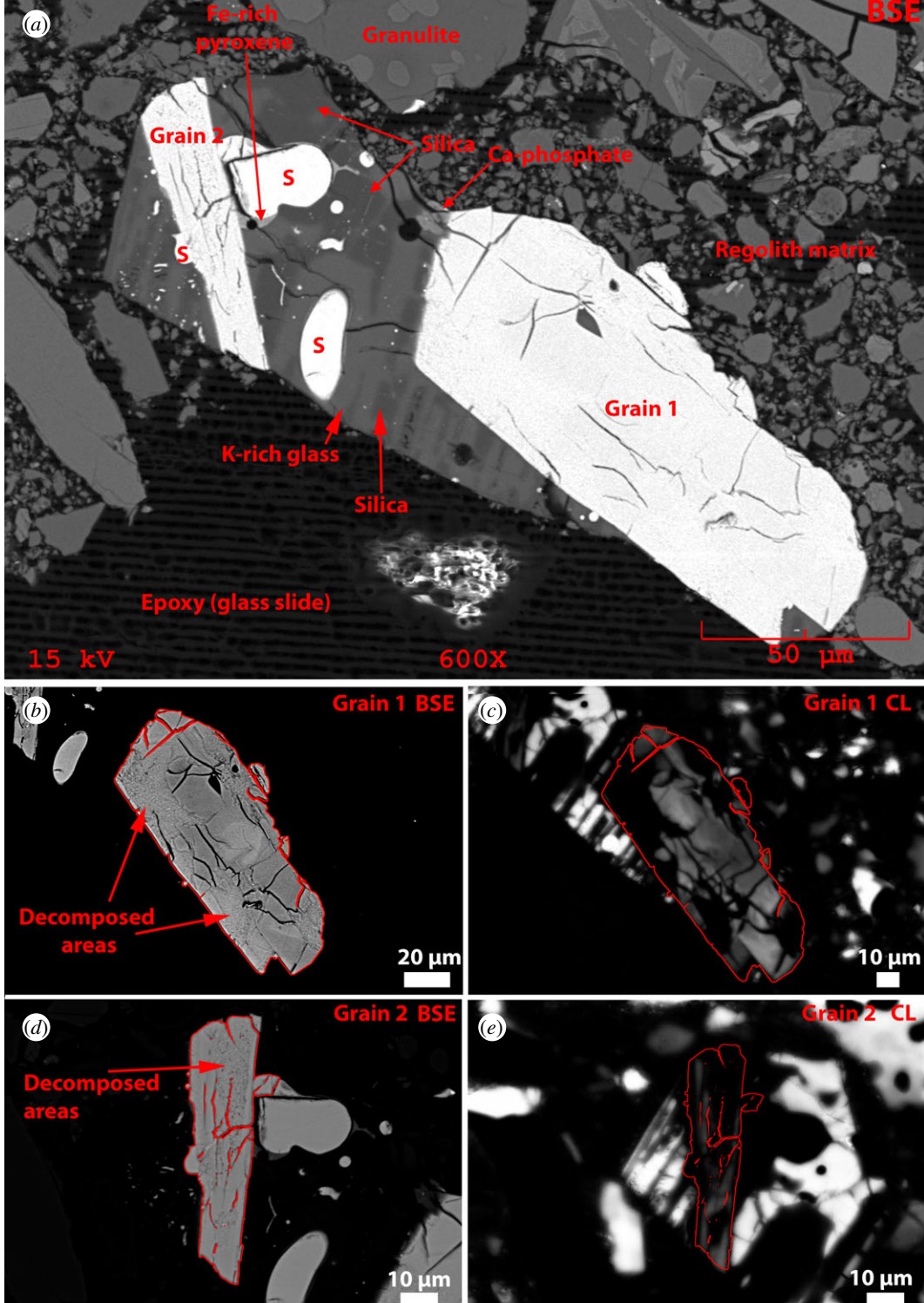

**Figure 3.** 65745,7 Clast 1. Granophyric assemblage hosting two zircon grains. (*a*) BSE image of the whole clast. S = sulfur. (*b*) Stretched contrast BSE image and (*c*) cathodoluminescence (CL) image of Grain 1 where the edge of grain and fractures are denoted by red lines. (*d*) Stretched contrast BSE image and (*e*) CL image of Grain 2 where the edge of grain and fractures are denoted by red lines.

16 kV. Analysis was preceded by *ca* 15 min presputtering of the area to eliminate any surface contamination. A first acquisition was carried out in peak-jumping mode to analyse $^{28}Si^+$, $^{48}Ti^+$, $^{56}Fe^+$, $^{89}Y^+$, $^{94}Zr^+$ and the Pb isotopes $^{204}Pb^+$, $^{206}Pb^+$, $^{207}Pb^+$ and $^{208}Pb^+$ (see electronic supplementary material, table S5). A complete cycle lasted *ca* 16 min 30 s, and 17 cycles were collected over almost 5 h. A second acquisition was then carried out for *ca* 2 h 20 min (32 cycles) over the same area to

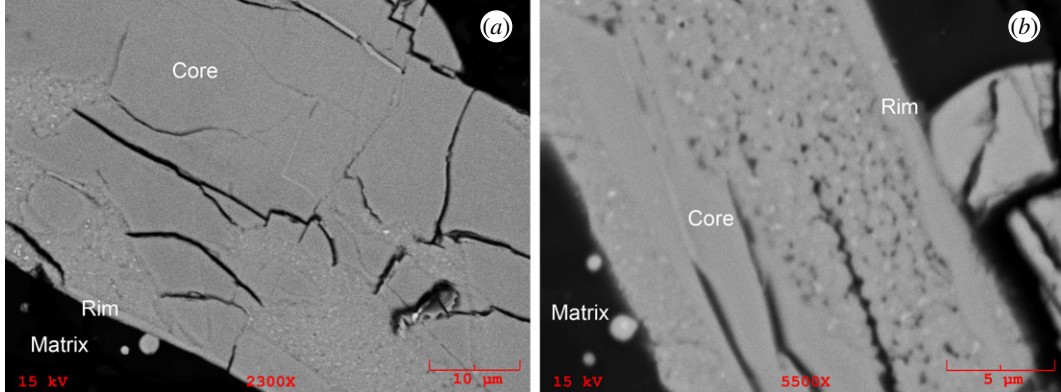

**Figure 4.** 65745,7 Clast 2. KREEP-rich impact melt breccia hosting one zircon grain and a plagioclase fragment. (*a*) BSE image of the whole clast. Plag = plagioclase. (*b*) Stretched contrast BSE image and (*c*) CL image of Grain 1 where the edge of grain and fractures are denoted by red lines.

**Figure 5.** Close-up BSE images of the Clast 1 (*a*) Grain 1 and (*b*) Grain 2 zircon showing details of the decomposed regions in these two grains. These images were collected prior to secondary ion mass spectrometry (SIMS) analysis. Decomposed areas have a speckled appearance with BSE bright and dark sub-micrometre-sized grains. Element maps of a similar region to (*a*) are available in figure 6 (see also figure 10).

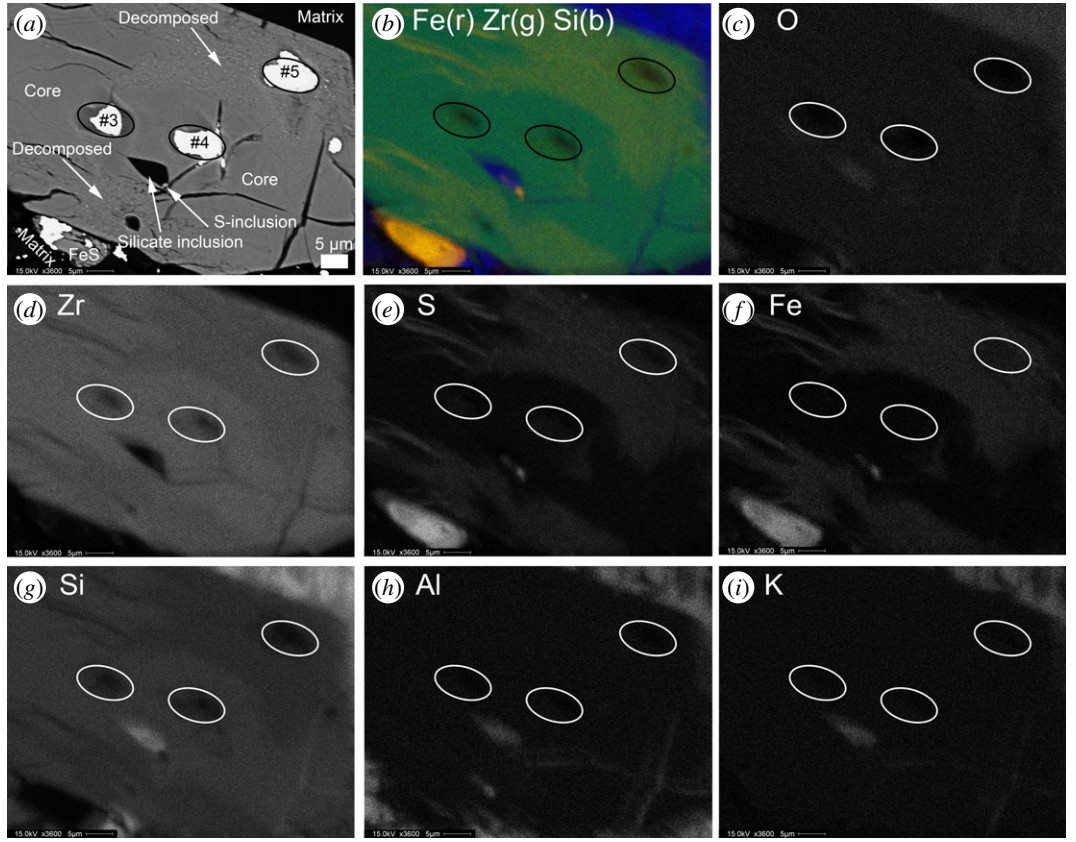

**Figure 6.** Clast 1 Grain 1 close-up BSE (*a*) and X-ray element distribution maps (*b*)–(*i*) of the zircon grain and decomposed areas. Locations of SIMS analytical spots #3, #4 and #5 (infilled with gold coat in (*a*)) are denoted. (*b*) False colour X-ray recombination map of Fe (red), Zr (green) and Si (blue) elemental distribution showing chemical variation in the grain between the zircon grain core and rim, and the two decomposed areas. Chemical composition variation between the core and decomposed areas is illustrated in (*c*) O, (*d*) Zr, (*e*) S, (*f*) Fe, (*g*) Si, (*h*) Al and (*i*) K individual element maps.

analyse the distribution of $^{28}Si^+$, $^{32}S^{16}O_2^+$, $^{232}Th^{16}O^+$ and $^{238}U^{16}O^+$. In order to adequately resolve isobaric interferences (e.g. HfSi molecules on Pb isotopes), the mass resolving power was set to approximately 6000 (CAMECA definition) using a 20 µm wide entrance slit (ES3) at the entrance of the mass analyser and a 200 µm wide aperture slit (AS2). The vacuum in the analysis chamber was approximately $7 \times 10^{-9}$ mbar. The data were processed off-line using the l'Image software package (L. Nittler, Carnegie Institution of Washington, Washington, DC) to produce the element distribution maps displayed in figure 7.

# 3. Results and interpretation

## 3.1. Petrography, mineral chemistry and cathodoluminescence results

Sample 65745 is a 7.76 gram, $2.6 \times 2.2 \times 1.2$ cm, regolith breccia collected to the south of the Apollo 16 landing site at Station 5 Stone Mountain [75] (electronic supplementary material, figure S1). The sample was described as a friable brownish soil breccia by Keil *et al.* [76], a type B3 (medium-coloured matrix, with light and dark clasts) by Wilshire *et al.* [77] and a soil-like regolith breccia by Jerde *et al.* [78] (electronic supplementary material, figure S2). The bulk rock composition [78] is consistent with other Apollo 16 regolith breccias and soil, indicating that it formed dominantly from feldspathic highlands material (ferroan anorthosites, feldspathic impact melts), with minor basaltic and KREEP chemical components (electronic supplementary material, figure S3).

Thin section 65745,7 (figure 2) is dark coloured with less than 0.8 mm angular-shaped clasts held in a fine-grained (less than 0.2 mm sized), loosely consolidated, clast-supported matrix (figure 2*a*). Clast types are mostly feldspathic impact melt breccias, rare mare basaltic components including low-Ti to high-Ti picritic volcanic glass beads (electronic supplementary material, figure S4), and mineral fragments. Impact melt spherules and agglutinates are distributed throughout the sample with a range of

**Table 1.** Average zircon and decomposed area chemical composition determined by EPMA at The University of Manchester (major and minor elements reported in electronic supplementary material, table S3) and SIMS (U, Th and Pb). Uncertainties are two standard deviations on the average measurement.

| | zircon | | | | | | decomposed area | | | |
|---|---|---|---|---|---|---|---|---|---|---|
| | Clast 1 | | | | Clast 2 | | Clast 1 | | | |
| | Grain 1 | | Grain 2 | | | | Grain 1 | | Grain 2 | |
| | $n = 14$ | $2\sigma$ | $n = 2$ | $2\sigma$ | $n = 6$ | $2\sigma$ | $n = 8$ | $2\sigma$ | $n = 3$ | $2\sigma$ |
| $SiO_2$ | 30.32 | ±0.44 | 30.54 | ±1.08 | 30.93 | ±0.51 | 26.04 | ±1.60 | 28.14 | ±1.64 |
| $TiO_2$ | | | | | <0.19 | | 0.12 | ±0.10 | 0.13 | ±0.05 |
| $Al_2O_3$ | | | | | <0.06 | | 0.17 | ±0.29 | 0.40 | ±0.35 |
| $FeO$ | 0.27 | ±0.27 | 0.45 | ±0.04 | 0.39 | ±0.34 | 11.82 | ±2.27 | 8.11 | ±1.97 |
| $MgO$ | | | | | <0.08 | | <0.04 | | 0.03 | ±0.01 |
| $CaO$ | 0.05 | ±0.04 | 0.28 | ±0.60 | 0.08 | ±0.07 | 0.18 | ±0.21 | 0.33 | ±0.55 |
| $K_2O$ | | | | | | | | | 0.15 | ±0.10 |
| $Sc_2O_3$ | | | | | 0.02 | ±0.01 | 0.03 | ±0.01 | | |
| $ZrO_2$ | 65.69 | ±0.73 | 66.22 | ±0.11 | 65.15 | ±1.13 | 54.23 | ±1.89 | 55.72 | ±2.91 |
| $ThO_2$ | | | | | | | 0.10 | ±0.02 | 0.10 | ±0.01 |
| $HfO_2$ | 1.33 | ±0.10 | 1.30 | ±0.14 | 1.16 | ±0.08 | 0.91 | ±0.05 | 1.02 | ±0.06 |
| $Y_2O_3$ | 0.17 | ±0.05 | 0.18 | ±0.06 | 0.84 | ±0.43 | 1.09 | ±0.37 | 1.28 | ±0.08 |
| total | 97.84 | | 99.07 | | 98.90 | | 94.62 | | 95.31 | |
| Hf/Zr (afu) | 0.014 | | 0.013 | | 0.011 | | 0.011 | | 0.013 | |
| Si/Zr (afu) | 0.949 | | 0.938 | | 0.974 | | 0.985 | | 1.036 | |
| SIMS | $n = 4$ | $2\sigma$ | | | $n = 3$ | $2\sigma$ | $n = 1$ | | $n = 2$ | $2\sigma$ |
| U (ppm) | 115 | ±35 | | | 165 | ±19 | 968 | | 805 | ±160 |
| Th (ppm) | 49 | ±18 | | | 176 | ±29 | 1425 | | 1192 | ±395 |
| Pb (ppm) | 165 | ±88 | | | 231 | ±26 | 1444 | | 1260 | ±256 |

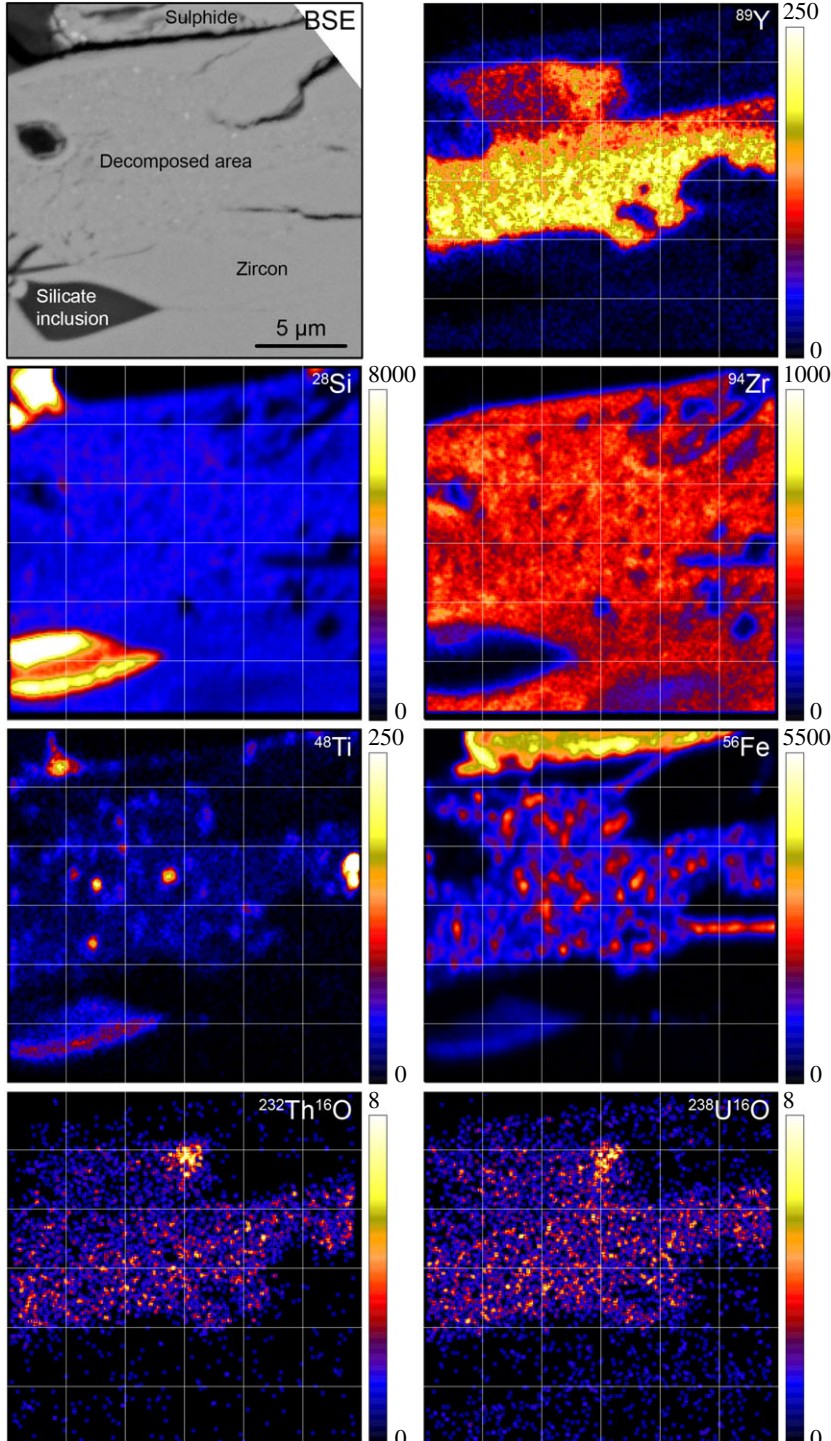

**Figure 7.** NanoSIMS distribution maps for selected species. The colour scale intensity corresponds to counts per second. A $3 \times 3$ pixel smoothing filter has been applied to all images.

compositions consistent with melting of both ferroan anorthositic parent lithologies, and more sodic varieties with an enhanced KREEP component (electronic supplementary material, figure S4c; table S2). We located two clasts in 65745,7 that host large zircon grains (figures 3 and 4).

### 3.1.1. Clast 1

Clast 1, approximately $200 \times 125$ µm is a micro-granitic assemblage with intergrown laths of K-rich glass (similar in composition to K-feldspar) and a silica phase (figure 3). The clast contains accessory rounded grains of troilite with sizes ranging from micrometre to sub-micrometre blebs up to *ca* 25 µm. Troilite is a

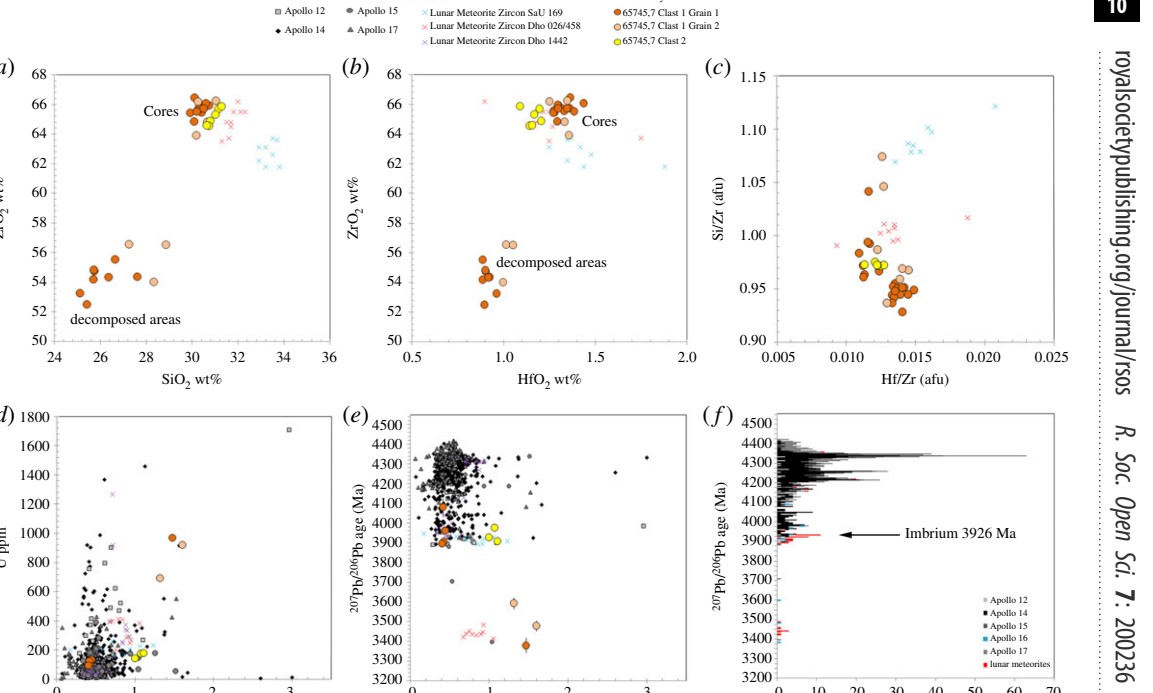

**Figure 8.** Composition and $^{207}$Pb/$^{206}$Pb dates of cores and decomposed regions for zircon grains in Clast 1 and 2 in 65745,7 (table 1; electronic supplementary material, table S3), compared with Apollo and lunar meteorite zircon data. $(a)$–$(c)$ major element composition as measured by EPMA. $(d)$–$(e)$ SIMS element data and $^{207}$Pb/$^{206}$Pb zircon dates. Apollo data from: A12: [11,58]; A14: [51,54,61,62]; A1: [5,6,62]; A17: [51,53–57,62]. Lunar meteorite data from granulitic feldspathic breccias Dho 026/458 [52,59], respectively; KREEPY regolith breccia Dho 1442 [60]; SaU 169 [58]. The age of the Imbrium impact basin-forming event is taken from Snape *et al.* [2] and references therein.

common accessory phase in lunar granitic lithologies [79]. EDS spectra also indicate the presence of small Ca-phosphate phases and a Fe-rich pyroxene (figure 3a).

Two large (50–150 µm) zircon grains are present in Clast 1 that we term Grain 1 and Grain 2. The core of the larger Grain 1 has linear and curviplanar fractures suggesting that low pressure less than 20 GPa dissociation and strain has occurred (figure 5a) [66,80,81]. These fractures are not filled with impact melt. This larger grain has a 5 × 8 µm silicate (Si-Al-K-rich) and an approximately 1 µm troilite inclusion, and is relatively chemically homogeneous with low-Y, low-Ti, low-U and low-Th abundances (figures 6 and 7; table 1). The zircon cores in the two grains are compositionally similar to each other and to other lunar zircon in terms of major elements (figure 8 and table 1). Both grains have irregular mosaicked CL textures (figure 3c and e), indicative of having seen shock deformation pressures in the 20 to 40 GPa range [82]. We do not observe any planar deformation features or lamellae of reidite (a high-pressure ZrSiO$_4$ polymorph).

BSE and CL images reveal that both zircon grains have decomposed regions where the crystal has broken down to form a polymineralic assemblage that has no cathodoluminescent response (figure 3c). The decomposed areas are the rim of Grain 1 and the core of Grain 2. Individual components of these domains are too small to determine their individual chemical composition using EDS mapping (figure 6). However, EPMA analysis of the decomposed regions and NanoSIMS imaging indicates that overall they have elevated FeO, TiO$_2$, Al$_2$O$_3$, REE and S, and lower ZrO$_2$, SiO$_2$ and HfO$_2$ contents compared with the crystalline zircon (table 1, and figures 6 and 8). We note that the analyses of the decomposed areas have low analytical totals of approximately 95 wt%, suggesting the presence of S, which was not determined by EPMA and yet shows up in EDS spectra and maps (figure 6), sub-micrometre pore spaces, and/or that multiphase sub-micrometre minerals domains cannot be accurately determined because of unequal host density effects [83]. High-resolution NanoSIMS mapping of a region of the Clast 1 Grain 1 decomposed zircon also shows that there seems to be two separate zones in the decomposed region—one richer in Y adjacent to the crystalline core, and a second one with lower Y abundance towards the zircon edge. NanoSIMS imaging also confirms that the decomposed region is enriched in Th and U compared with the crystalline zircon core (figure 7). In BSE images, the decomposed assemblage has a speckled texture appearance with sub-micrometre BSE bright and dark domains (figures 3a and 5). NanoSIMS imaging

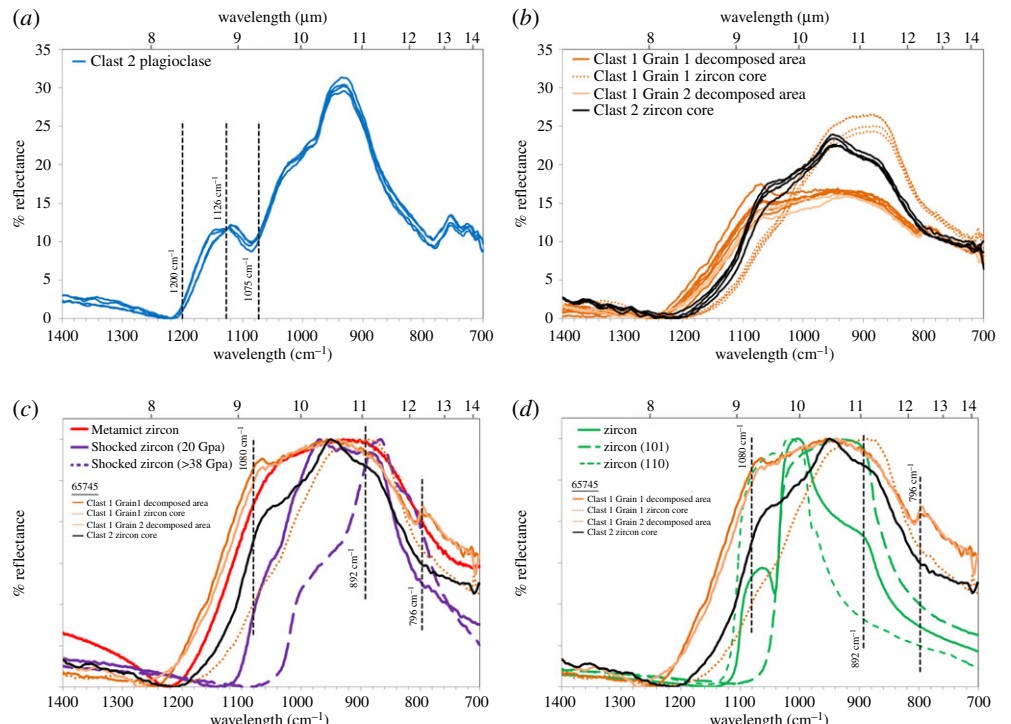

**Figure 9.** Unpolarized IR reflectance (in %) spectra normalized to minimum reflectance between 700 and 1400 cm$^{-1}$ for plagioclase and zircon in Clast 1 and 2 65745,7 (electronic supplementary material, table S4). (*a*) Plagioclase fragment in the Clast 2 impact melt matrix (figure 4*a*). The band positions of the 1126 cm$^{-1}$ band depth shock indicator is denoted (see text and [8] for details of calculation). (*b*) All spectra collected in 65745 Clast 1 and Clast 2 zircon phases including cores and decomposed areas. (*c*) Representative spectra from the 65745 Clast 1 and Clast 2 zircon phases compared with (i) spectra of literature shocked zircon where the purple colour lines are the 20 GPa (non-reidite bearing) and 38–80 GPa (reidite bearing) experiments of Gucsik *et al.* [84], and (ii) a Sri Lankan metamict zircon (Caltech Mineral Spectroscopy Server: http://minerals.gps.caltech.edu/index.html). (*d*) Representative spectra from the 65745 clast 1 and clast 2 zircon phases compared with green-coloured spectra of unshocked zircon in 101 (polarized data) and 110 (polarized data) orientations, and mixed orientation (Zircon, grr 3299, Singida, Tanzania: Caltech Mineral Spectroscopy Server). Spectra in (*a*) and (*b*) have been normalized to their lowest reflectance % value. For cross comparison, spectra in (*c*) and (*d*) have been doubly normalized to their lowest reflectance value and then also to their highest reflectance value between 1400 and 700 wavenumber. Diagnostic bands are indicated as vertical dashed lines as discussed in the text.

suggests that the bright phases appear to be Fe-rich and Ti-rich micrometre scale to sub-micrometre domains, whereas the dark areas appear to be small holes in the sample surface rather than small Si-rich phases (figure 7).

### 3.1.2. Clast 2

Clast 2 is composed of a microcrystalline vermicular glass that was probably formed as a rapidly quenched impact melt. This glassy matrix has a bulk composition (electronic supplementary material, table S1) that is as sodic and ferroan as HAS rocks (electronic supplementary material, figure S4c), with $Al_2O_3$ and FeO abundances intermediate between those of KREEP and mare basalts (electronic supplementary material, figure S3c and S3d). The clast contains a fragment of anorthite ($An_{90-91}$, $Al_{8-9}$, $Or_1$: electronic supplementary material, table S1) with a smooth interior lacking fractures, and resorbed edges that look like they have been partially melted/resorbed by the impact melt matrix (figure 4*a*). The *ca* 50 µm long zircon grain appears homogeneous when examined using BSE, but displays different CL domains including dark and bright bands approximately 3–6 µm thick (figure 4*c*) that could be shock-induced planar deformation bands consistent with shock pressures in the less than 20 GPa range [81].

### 3.2. Fourier transform infrared results

We measured the non-polarized mid-IR reflectance spectra of both the core and decomposed regions of zircon grains in Clast 1, of the Clast 2 zircon, and of plagioclase grains (figure 9; electronic supplementary material, table S4).

### 3.2.1. Plagioclase

The FTIR spectra of the Clast 2 plagioclase are typical of anorthitic plagioclase that has undergone some degree of shock modification [86]. The band depth at 1126 cm$^{-1}$, after removal of the continuum between 1075 and 1200 cm$^{-1}$, can be used to quantify shock pressure effects (see method of [8]). Application of this calculation suggests that the plagioclase experienced between 13 and 17 (±3) GPa of shock pressure, which is below the greater than 20–24 GPa shock pressure boundary needed to convert anorthitic plagioclase to diaplectic glass [86,87]. This puts an upper limit on the shock pressures witnessed by Clast 2 after it was formed as an impact melt fragment.

### 3.2.2. Zircon

Reflectance IR spectra in zircon are strongly affected by crystal orientation effects (i.e. [89]), and bands between 1010 and 933 cm$^{-1}$ and at 892 cm$^{-1}$ relate to internal stretching modes of the $[SiO_4]^{4-}$ anion ([91] and references therein) in the zircon crystal structure. FTIR reflectance and absorption measurements have previously been used to understand modification of vibration bonds in zircon caused by (i) high pressure, low temperature, shock damage (e.g. [84,91]), (ii) metamictization from radiation damage (e.g. [89,91,92]), and (iii) thermal annealing effects from high temperature shock or contact metamorphism [88].

All the zircon phases analysed in sample 65745,7 have bands in the spectral region between 1100 and 800 cm$^{-1}$ resulting from the fundamental vibration modes of zircon Si-O-Zr bonds, although there are appreciable differences in terms of reflectance strength and band position between the (i) Clast 1 zircon cores, (ii) the Clast 1 decomposed regions (Grains 1 and 2), and (iii) the Clast 2 zircon.

Spectra of the decomposed regions of Clast 1 were collected using 6.25 µm spot sizes, and so represent a bulk measurement of this complex sub-micrometre-grained scale domain. These areas have reflectance spectra that are a third less reflective than the zircon cores, with a distinctive band at approximately 796 cm$^{-1}$ and variable magnitude bands around approximately 1074 cm$^{-1}$. They also have higher average Christianson feature (CF) positions (i.e. the point of minimal reflectance) of 1252–1273 cm$^{-1}$ compared with Clast 1 Grain 1 (1215 cm$^{-1}$) and the Clast 2 zircon (1223 cm$^{-1}$). Generally, the loss of intensity of bands results from radiation damage (i.e. metamictization: [89]) or shock in the 20 GPa pressure range [84] caused by structural distortion and strain. Both of these processes will cause internal vibration bonds to weaken in reflectance strength and broaden in wavenumber, indicating that Si-O-Zr bonds in the $ZrSiO_4$ group in the crystal lattice have been damaged to form shorter Si–O bonds.

The crystalline cores of the Clast 1 and Clast 2 zircon have reflectance spectra that are broader and have less obvious bands than unshocked terrestrial zircon (figure 9d). The core of Clast 1 Grain 1 has a broad and flattened reflectance band from 892 to 950 cm$^{-1}$ with a minor band at approximately 1060 cm$^{-1}$, but is shifted to lower wavebands than Clast 2 and the unshocked terrestrial zircons. This characteristic shift and flattening of the main band from 892 to 950 cm$^{-1}$ is similar to that seen in experimental shocked zircons in the approximately 20 GPa pressure range [84] (figure 9c). Taken with the CL observations, this suggests that the Clast 1 zircon grain core underwent shock at approximately 20 GPa. In comparison, the Clast 2 zircon has distinct reflectance bands at 892, 950, and 1060 cm$^{-1}$, which have not been shifted to lower wavenumbers as seen in shock experiments [84]. While we do not consider this zircon grain as completely unshocked as it has a broader reflectance spectrum than unshocked terrestrial zircon (figure 9d), we estimate that the grain cores in Clast 2 did not witness shock above approximately 19 GPa. This is consistent with the Clast 2 maximum plagioclase shock pressure regime of 13–17 GPa (see above).

## 3.3. Secondary ion mass spectrometry results

### 3.3.1. Clast 1

We carried out seven SIMS analyses in Clast 1 (figures 8 and 10): five in Clast 1 Grain 1 and two in Clast 1 Grain 2 (table 2). Two of the analyses from the core of Grain 1 are concordant (0.8 to 2.7% discordance) and yielded $^{207}Pb/^{206}Pb$ dates of 3905 ± 14 Ma (analysis #1) and 3896 ± 17 Ma (analysis #4), and $^{206}Pb/^{238}U$ dates of 3984 ± 183 Ma (analysis #1) and 3918 ± 193 Ma (analysis #4). Combined, these two data points give a weighted mean $^{207}Pb/^{206}Pb$ date of 3901 ± 11 Ma (2σ) with a mean square weighted deviation (MSWD) = 0.67 and probability = 0.41.

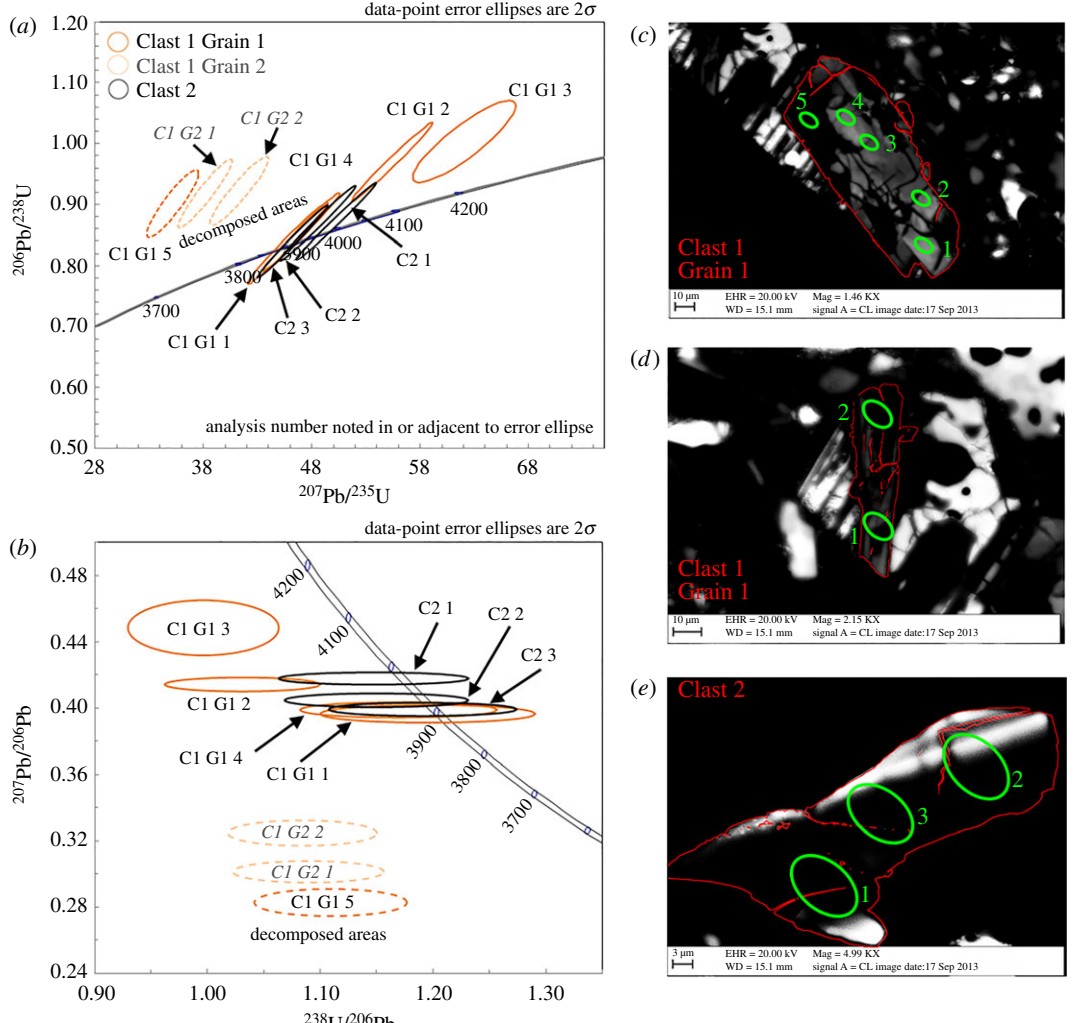

**Figure 10.** (a) and (b) are U-Pb isotope concordia plots. Data shown have been corrected for initial Pb using the modern Stacey & Kramers [74] composition. Error ellipses are given at the 2σ level. In (c), (d) and (e), the locations of the SIMS analysis spots have been overlaid on the CL images shown in figure 3. Zircon grain outlines are shown in red, SIMS spots in green.

Of the seven analyses, five are reversely discordant (figure 10). Two analyses with discordance at the 10% level were made where the SIMS spots were positioned on a crack in the core of the Clast 1 Grain 1 zircon; surface relief variations or voids associated with the crack may be responsible for the reverse discordance of the U-Pb systematics, but should not affect Pb isotope ratios, which are not easily fractionated during SIMS analysis (e.g. [93]), where the accuracy of $^{207}$Pb/$^{206}$Pb dates depends only upon reliably estimating the common Pb composition: these yield apparent $^{207}$Pb/$^{206}$Pb dates of $3962 \pm 12$ Ma (analysis #2) and $4080 \pm 45$ Ma (analysis #3). Data with higher levels of discordance (e.g. 24–31%) were acquired in the decomposed areas; we attribute such high levels of reverse discordance to issues with the U-Pb matrix-match standardization for such multi-domain phases (i.e. the standards used are crystalline zircon grains, as opposed to decomposed areas). Similarly to the possible influence of a crack on U-Pb discordancy discussed above, Pb isotope ratios should remain accurate, despite this matrix mismatch. The $^{207}$Pb/$^{206}$Pb apparent dates obtained on these decomposed areas are $3379 \pm 37$ Ma (Grain 1 analysis #5), $3593 \pm 30$ Ma (Grain 2 analysis #1) and $3478 \pm 27$ Ma (Grain 2 analysis #2).

In spite of their discordant older apparent $^{207}$Pb/$^{206}$Pb dates, the four SIMS spots in the core of Grain 1 all have similar U (95 to 133 ppm), Th (38 to 60 ppm) and Pb (117 to 205 ppm) concentrations, similar to other Apollo sample zircon grains (figure 8d–f and table 2). The decomposed areas all have elevated U (692 to 968 ppm), Pb (1079 to 1444 ppm) and Th (913 to 1471 ppm) abundances compared with the zircon core and with Apollo zircons in general, suggesting mobility of these elements. This observation is consistent with NanoSIMS observations of elevated U, Th and Pb in the decomposed area compared with the crystalline zircon areas (figure 7).

**Table 2.** U-Pb and Pb-Pb data from Zr-rich phases in 65745,7. Uncertainties on the dates are reported to $1\sigma$ in this table and are discussed at $2\sigma$ level in the text. All tabulated uncertainties are $1\sigma$. (1) Based on measured Th and U signals; (2) $f_{206}$ is the percentage of common Pb signals; (3) ratios after subtraction of common Pb estimated from $^{204}$Pb counts; (4) discordance in %, values in parentheses indicate concordant within $2\sigma$ uncertainty.

| sample ID | texture | concentrations (ppm) | | | $Th/U^1_{meas}$ | $^{206}Pb/^{204}Pb$ | $f_{206}$ (%)[2] | radiogenic ratios[3] | | | | dates ± 1$\sigma$ (Myr) | | disc. (%)[4] |
| | | U | Th | Pb | | | | $^{206}Pb/^{238}U$ | ±σ (%) | $^{207}Pb/^{206}Pb$ | ±σ (%) | $^{207}Pb/^{206}Pb$ | $^{206}Pb/^{238}U$ | |
| --- | --- | --- | --- | --- | --- | --- | --- | --- | --- | --- | --- | --- | --- | --- |
| Clast 1, Grain 1, Spot 1 | crystalline core | 110 | 46 | 140 | 0.43 | 6640 | 0.28 | 0.8551 | 3.0 | 0.3990 | 0.48 | 3905 ± 7 | 3984 ± 91 | [2.7] |
| Clast 1, Grain 1, Spot 2 | crystalline core | 130 | 60 | 200 | 0.45 | 4320 | 0.43 | 0.9699 | 2.7 | 0.4143 | 0.42 | 3962 ± 6 | 4371 ± 87 | 14.3 |
| Clast 1, Grain 1, Spot 3 | crystalline core | 120 | 52 | 200 | 0.42 | 6700 | 0.28 | 1.0036 | 2.7 | 0.4486 | 1.5 | 4080 ± 22 | 4480 ± 89 | 13.6 |
| Clast 1, Grain 1, Spot 4 | crystalline core | 95 | 38 | 120 | 0.41 | 4120 | 0.45 | 0.8366 | 3.3 | 0.3966 | 0.55 | 3896 ± 8 | 3919 ± 96 | [0.8] |
| Clast 1, Grain 1, Spot 5 | decomposed | 970 | 1400 | 1400 | 1.5 | 21 600 | 0.09 | 0.9016 | 2.5 | 0.2829 | 1.2 | 3379 ± 18 | 4143 ± 77 | 30.9 |
| Clast 1, Grain 2, Spot 1 | decomposed | 690 | 910 | 1100 | 1.3 | 11 700 | 0.16 | 0.9225 | 2.5 | 0.3248 | 0.97 | 3593 ± 15 | 4213 ± 78 | 23.7 |
| Clast 1, Grain 2, Spot 2 | decomposed | 920 | 1500 | 1400 | 1.6 | 17 000 | 0.11 | 0.9179 | 2.5 | 0.3014 | 0.88 | 3478 ± 14 | 4198 ± 78 | 28.4 |
| Clast 2, Spot 1 | crystalline core | 170 | 190 | 250 | 1.1 | 29 300 | 0.06 | 0.8715 | 3.0 | 0.4179 | 0.36 | 3975 ± 5 | 4040 ± 91 | [2.2] |
| Clast 2, Spot 2 | crystalline core | 140 | 140 | 200 | 1.0 | 8510 | 0.22 | 0.8697 | 2.9 | 0.4048 | 0.42 | 3927 ± 6 | 4034 ± 87 | [3.7] |
| Clast 2, Spot 3 | crystalline core | 180 | 200 | 240 | 1.1 | 47 000 | 0.04 | 0.8398 | 2.9 | 0.3993 | 0.41 | 3906 ± 6 | 3930 ± 85 | [0.8] |

### 3.3.2. Clast 2

Three SIMS analyses were made in zircon in Clast 2. All three U-Pb dates are concordant, although we note that analysis #1, which corresponds to an older $^{207}$Pb/$^{206}$Pb date of 3975 ± 11 Ma, was located on a small crack in the grain (figure 10). The other two analyses yielded $^{207}$Pb/$^{206}$Pb dates of 3927 ± 13 Ma (analysis #2) and 3906 ± 12 Ma (analysis #3) that are consistent with the two concordant dates obtained in Clast 1 Grain 1. However, the chemical composition of zircon in Clast 2 is somewhat different to that of the cores of Clast 1 zircon grains, having higher U (143–179 ppm), Pb (201–248 ppm) and Th (143–198 ppm) concentrations, and elevated Th/U ratios (0.99–1.1) compared with the core of Clast 1 zircon (Th/U 0.4–0.45: figure 8d–f and table 2).

## 4. Discussion

This study provides new U-Pb geochronological data for zircon from Apollo 16 samples. The $^{207}$Pb/$^{206}$Pb dates reported here have been corrected for $^{204}$Pb terrestrial contamination effects and should not be affected in the same way as inter-element ratios (i.e. U/Pb ratio dates) by surface imperfections (i.e. cracks) or matrix-match standardization issues (e.g. the decomposed areas of the zircons). Our discussion about the significance of the measured dates will, thus, focus only on these $^{207}$Pb/$^{206}$Pb dates.

### 4.1. Clast 1 origin and evolution

The clast is geochemically evolved and reminiscent of granophyric textures reported in lunar samples. We interpret that the zircon grains and other mineral phases (K-glass, silica, sulfides, Fe-pyroxene) incorporated within Clast 1 were all part of the same granitic rock assemblage. However, it is challenging to interpret the clast's original petrogenesis: it may be a primary fine-grained HAS intrusive or extrusive rock (akin to those reported by [50,79]) that formed magmatically during extreme melt fractionation or through silicate liquid immiscibility [79,94]. Alternatively, the fragment may be a low-pressure quenched rock derived from impact-driven partial melting of a HAS precursor.

The oldest Pb-isotope date recorded in the crystalline zircon core in this clast is 4080 ± 22 Ma, taken to represent the latest time of zircon crystallization. This date could reflect the timing of the clast's magmatic age, or a significant metamorphic isotopic resetting event. Compared with the dates obtained on other zircon grains in the Apollo collection (figure 8c,d), this 4.08 Ga date is younger than the approximately 4.33 Ga spike in zircon $^{207}$Pb/$^{206}$Pb dates associated with ancient lunar magmatic events [61]. It is also younger than the dates of ca 4.2 Ga obtained for the formation or isotopic resetting dates of some Apollo 16 impact melt samples [40,47,49,95–97]. As such, we interpret the 4080 Ma date obtained in 65745,7 zircon core as most likely representing an intermediate, partial resetting of the zircon U-Pb system. The two concordant analyses with a weighted mean $^{207}$Pb/$^{206}$Pb date of 3901 ± 11 Ma obtained in regions of the crystalline zircon core are similar to resetting ages caused by thermal heating during emplacement of Cayley Plains ejecta blanket when the Imbrium basin was emplaced across the nearside of the Moon (figure 1) at 3926 Ma [2,58]. Our FTIR results (figure 9) suggest that the crystalline zircon cores have been shocked to approximately 20 GPa. We interpret that the clast was included within the distal Imbrium basin ejecta blanket at this time and was partially reset either by this basin-forming event or by other large nearside impacts at this time [98].

The decomposition texture in parts of Clast 1 zircon grains suggests that resetting of the Pb isotope system occurred at approximately 3.4 Ga. The scatter of these analyses indicates partial resetting so the youngest date (3379 Ma) is a maximum age estimate for the resetting event. This is one of the youngest $^{207}$Pb/$^{206}$Pb dates recorded in an Apollo zircon. Below we explore some of the possibilities through which these decomposed areas could have formed.

### 4.1.1. Thermal effects?

In Clast 1, grain boundaries between the K-rich glass and the silica phase are diffuse, and sulfide grains are rounded (figure 3a); it is clear that the rock fragment suffered from a post-crystallization thermal pulse at, or close to, the system solidus to be able to anneal the clast's silicate and sulfide mineral phases. This is thought to be about 990°C for lunar felsic granophyre systems [79]. It is possible that this thermal pulse could have partially thermally annealed the zircon grain, causing partial decomposition, inducing Pb, U and Th mobility, and exceeding the closure temperature of zircon (approx. 900°C) to cause resetting of the U/Pb isotopic systematics.

### 4.1.2. Shock effects?

Previously analysed zircon crystals from the Apollo 16 site appear to have witnessed relatively low shock pressures of mostly less than 20 GPa and always less than 40 GPa [66]. The decomposed zircon grains described here are somewhat more texturally similar to the highly shocked grains reported by Zhang *et al.* [59] in lunar meteorite Dhofar 458 and by [5,6] in Apollo 15 impact melt 15405,145 (Clast M3). We note that the *ca* 3.4 Ga $^{207}$Pb/$^{206}$Pb date recorded by the decomposed regions of zircon in 65745,7 is also similar to 3.4 Ga dates recorded by these shocked zircons (figure 8*e* and *f*). However, in comparison with these grains, the 65745,7 Clast 1 decomposed areas have much higher U and Pb contents (692–968 ppm U, 913–1425 ppm Pb, compared with 67–174 ppm U, 70–164 ppm Pb in 15405,5 Clast M3, and 201–414 ppm U, 151–406 ppm Pb in Dhofar 458), suggesting Pb and U were gained during the zircon breakdown process, rather than loss of Pb typically expected during resetting of the U-Pb system during a shock event [99,100]. Moreover, FTIR reflectance spectra from the 65745,7 Clast 1 decomposed areas are dissimilar to zircon crystals shocked experimentally between 38 and 80 GPa where reidite (high-pressure Zr-polymorph) has been observed to form under low temperatures [80,85] (figure 9*c*). We also do not observe nanoscale baddeleyite or Si-phases in the decomposed area (figure 7) indicating that the transition from zircon to Zr-polymorphs+silica dissociation has not occurred on the sub-micrometre scale (figures 6*g* and 7). Taken together, the lack of CL features (figure 3*c* and *e*), lack of reidite (figure 9) and the micro-domain texture of the decomposed areas (e.g. Figure 5) could indicate that portions of the grain were shocked non-uniformly above 80 GPa to cause near-complete to complete structural breakdown [82,84].

### 4.1.3. Radiation damage?

Alternatively, the decomposed areas could have formed in response to radiation damage caused by radioactive decay of naturally occurring radionuclides and their daughter products in the $^{238}$U, $^{235}$U and $^{232}$Th decay series (see study of a lunar zircon by [101]). A broad peak in FTIR spectra from 796 to 1080 cm$^{-1}$ and the appearance of reflectance peaks at 796 cm$^{-1}$ and 1100 cm$^{-1}$ in the decomposed areas are consistent with such types of damage (i.e. mixtures of amorphous and crystalline phases) caused by 'moderate' radiation levels or thermal annealing of grains above 727°C [88,89,91,92].

Using the U, Th and Pb abundances measured in the zircon crystalline and decomposed areas (table 2), we can calculate potential radiation doses ($\alpha$-events g$^{-1}$) for different periods of time between 4.1 and 3.4 Ga. Regardless of the age decay model used, the U, Th and Pb abundances in the zircon cores indicate doses less than $0.5 \times 10^{16}$ events mg$^{-1}$, consistent with crystalline grains that have not undergone radiation damage. If the zircon was self-shielded for 500 Ma from 3.9 until 3.4 Ga, calculated dose rates of between $0.4–0.55 \times 10^{16}$ events mg$^{-1}$ that account for the U, Th and Pb abundances in the decomposed areas, also do not indicate the likelihood of appreciable accumulating radiation damage. To induce significant annealing and metamictization of the zircon mineral structure, we would have to invoke a much longer self-shielding period of a 900 Ma decay interval from 4.3 until 3.4 Ga (i.e. doses of $0.8–1.1 \times 10^{16}$ events mg$^{-1}$).

Comparing these options, we do not think that radiation damage is likely to have contributed significantly to the zircon decomposition. It seems more likely that a combination of high temperature (greater than 900°C) effects and non-uniformly distributed localized shock-induced pressure effects (greater than 80 GPa) during an impact event probably contributed to the mineral structure breakdown, element diffusion and isotopic resetting.

## 4.2. Clast 2

Clast 2 is a clast-bearing impact melt breccia with a fine-grained quench textured matrix that is KREEP-rich and chemically similar to rocks from the Apollo HAS (electronic supplementary material, figures S3 and S4). We interpret that the mineral grains were mixed into the glass during an impact melt event, rather than being formed *in situ* from the melt itself. The occurrence of fine-grained quench glass indicates whole rock shock melting of a felsic silicate mineral system at high temperatures (greater than 1200°C) and pressures (greater than 60 GPa: [87]). However, the mineral grains in the breccia (plagioclase and zircon) have witnessed relatively low shock pressures (zircon: less than approximately 19 GPa; An$_{90–91}$ plagioclase 13 to 17 GPa: figure 9) and have not been significantly melted (apart from some incipient melting of grain boundaries, notably around the edge of the plagioclase). The zircon grain preserved as a clast in the breccia is chemically distinct from the zircon

seen in Clast 1 (figure 8) (higher Th/U ratios, lower $HfO_2$), suggesting a different petrological origin. The $^{207}Pb/^{206}Pb$ dates obtained on the Clast 2 zircon grain (3927 ± 13 to 3906 ± 12 Ma) are within error of the U-Pb age of the Imbrium basin-forming event at 3926 Ma [2,58].

# 5. Conclusion

Based on the petrography, geochemical characteristics, IR spectroscopy and zircon U-Pb and Pb-Pb isotope systematics of two zircon-bearing clasts in Apollo 16 regolith breccia sample 65745,7, we propose the following geological history for these clasts:

## 5.1. Pre-Imbrium events

The clasts have mineralogical and chemical affinities to evolved magmatic parent rocks. However, it seems likely, given their different petro-geochemical characteristics (figure 8), that these two clasts were not derived from the same petrological source. We note that Clast 1 is mineralogically akin to the silica-rich type of HAS samples, which are rare at the Apollo 16 landing site (most of these rocks have been recovered from Apollo 12, 14, 15 and 17). The U-Pb and $^{207}Pb/^{206}Pb$ dates obtained on Zr-rich minerals in Apollo evolved magmatic rocks (i.e. the HAS) range from 4.34 to 4.03 Ga for alkali anorthosites, and 4.32 to 3.97 Ga for silica-rich, K-feldspar-rich (granite-like and quartz monzogabbro) samples ([46] and references therein; [51,102,103]). It remains unclear whether the $^{207}Pb/^{206}Pb$ dates obtained in the crystalline areas of the studied zircon grains reflect a crystallization age for their parental evolved magmas, or reflect a single or multiple isotopic impact reset ages at approximately 4.1 Ga (Clast 1 zircon) and 3.9 Ga (Clast 1 and Clast 2 zircon).

## 5.2. Effects of the Imbrium basin-forming event

It seems plausible that the *ca* 3.9 Ga dates of Clast 1 and Clast 2 zircon reflect partial to total isotopic U/Pb-isotope resetting by the Imbrium basin-forming event [47,58,104–106]. This event excavated into the Moon's upper nearside crust, extracting many different rock types including HAS magmatic intrusions, mixing them together as clastic loads that were carried within the Cayley Plains ejecta blanket emplaced as continuous ejecta at the Apollo 16 landing site (figure 1) (see [39] and references therein). However, preliminary calculations [107] suggest Imbrium impact ejecta deposited at a distance of 1600 km (i.e. at the Apollo 16 site) would retain average temperatures of only approximately 170°C for hundreds to a few thousand years after the impact (assuming a warm lunar thermal model and ejecta thicknesses of 75 to 205 m: [39]), which is not above the zircon closure temperature (greater than approx. 900°C). It may be that higher thermal pulses occurred during the excavation stage of the basin-forming process when the clasts/zircons were excavated from depth, or that thermal resetting occurred when the clasts/zircons were included within higher temperature impact melt components of the Imbrium ejecta blanket.

## 5.3. Disturbance at 3.4 Ga

The zircon grains sampled within Clast 1 suffered a later period of disturbance at approximately 3.4 Ga. This may have been a thermal and/or irregularly distributed shock pressure pulse, which facilitated decomposition of the zircon grains and caused silicate and sulfide minerals to be thermally annealed. This event caused U, Pb and Th mobility within the zircon, concentrating these elements within the decomposed zones. The timing of this resetting event is consistent with a window of isotopic resetting events between 3.4 and 3.5 Ga witnessed by the U-Pb system in shocked zircon grains in feldspathic lunar meteorite Dhofar 458 [59] and Apollo 15 impact melt breccias [5,6], and by Ar/Ar dates recorded in Apollo 16 [40,108,109] samples, Apollo 17 samples [110], and in feldspathic lunar meteorites [111–114]. This implies the possible occurrence of post-basin formation crater formation episodes occurring across both the lunar nearside (Apollo) and farside highlands regions (feldspathic lunar meteorites) throughout the Imbrian period.

## 5.4. Regolith breccia formation

After Clast 1 and Clast 2 formation, excavation from their parent magmatic rocks by the Imbrium basin-forming event, transport to the Apollo 16 landing site as part of the Cayley Plains at *ca* 3.9 Ga, and likely

disruption by a shock event at approximately 3.4 Ga, they were at some stage liberated from the Imbrium ejecta blanket and worked into the local regolith. This 65745 parent soil is submature with a maturity indicator $I_s/FeO = 27$ [78], suggesting that it had experienced moderate amounts of time at the lunar surface (approx. timescale of tens of millions of years of space exposure). The 65745 regolith breccia is classified as 'soil-like' in nature, suggesting that it is a young breccia that probably consolidated from a soil into a breccia in the last 1 billion years or so [39,115], encapsulating the two clasts into their current rock assemblage (figure 2) prior to being collected close to Stone Mountain by the Apollo 16 astronauts in 1972 (electronic supplementary material, figure S1 [116]).

Data accessibility. Data have been uploaded as part of the electronic supplementary material.
Authors' contributions. K.H.J. conceived the study and wrote the manuscript. J.F.S., A.A.N. and M.J.W. undertook the SIMS analysis. D.M.M. and J.F.P.-F. assisted with the FTIR data collection and interpretation. R.T. assisted with NanoSIMS data collection. V.V. assisted with the CL data acquisition. D.A.K. supported the regolith breccia research. All authors have read and have contributed to the final manuscript.
Competing interests. We declare we have no competing interests.
Funding. This research was facilitated by STFC (PhD studentship to Dayl Martin, and grant nos. ST/M001253/1, ST/P005225/1 and ST/L002957/1) and the Royal Society (grant no. RS/UF140190) funding. K.H.J. acknowledges NASA Lunar Science Institute contract NNA09DB33A to D.A.K., which supported the JSC EMP analyses and thin section mapping. This is LPI Contribution number 2366. A.A.N. and M.J.W. were co-PI's on the Knut and Alice Wallenberg Foundation grant no. 2012.0097 and Swedish Research Council grant no. 2012–04370, and the NordSIMS laboratory was funded under Swedish Research Council grant no. 2014–06375; this is NordSIMS contribution number 640. J.F.S. acknowledges funding from the European Commission Horizon 2020 Research and Innovation programme (Marie Skłodowska-Curie Actions Fellowship grant no. 794287). At The University of Manchester, the NanoSIMS was funded by UK Research Partnership Investment Funding (UKRPIF) Manchester RPIF Round 2.
Acknowledgements. K.H.J. would like to thank the LPI library staff for digitizing the Warner et al. Apollo 16 technical report, and George Rossman at the Caltech Mineral Spectroscopy Server and Ming Zhang for their help locating zircon FTIR data. Also thanks to NASA JSC ARES staff for Roger Harrington for help with the thin section. We also thank Dr Marc Norman and two anonymous reviewers for their helpful comments on the manuscript.

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
