## [Reviewer comments · Royal Society Open Science]

Review History

RSOS-200236.R0 (Original submission)

Review form: Reviewer 1

Is the manuscript scientifically sound in its present form?

Yes

Are the interpretations and conclusions justified by the results?

Yes

Is the language acceptable?

Yes

Do you have any ethical concerns with this paper?

No

Have you any concerns about statistical analyses in this paper?

No

Recommendation?

Accept with minor revision (please list in comments)

Comments to the Author(s)

This study investigated 2 Apollo 16 clasts which contain zircon. Clast 1 has a zircon interpreted to be from a high alkali suite rock, with zircon ages that range between 4.01 and 3.4 Ga. Clast 2 U-Pb ages are between 3.97 and 3.9 one Ga and is more akin to KREEP-rich impact melt breccia. Data were collected by in situ ion microprobe. Overall the authors present a nice description of the data which also includes spectroscopy measurements and trace element analysis. The microchemical features of the grains are well described, as are the physical features including curvy planar features indicating what the authors interpret to be low pressure <20 GPa impact shocking. This is a nice contribution and will be welcome by the community and I recommend publication. With this said I do raise several considerations for the authors to address or respond to before this manuscript is published.

1) L109 The authors had provided insufficient detail to evaluate the EMPA rare earth element measurements that have been conducted on zircon. The elements that the authors have targeted by electron microprobe are extremely difficult measurements with multiple X-ray interferences and overlaps. Moreover the overlaps between the rare earth element doped glass and the zircons may not be systematic. That is, the rare earth elements doped glasses could be doped uniformly (i.e., not with relative chondrite normalized abundances) and may not therefore reflect the "odd even" effects for natural samples. While I would not normally request this for an electron microprobe measurement these measurements are so difficult that I think the authors need to provide the location of their backgrounds and the location of their peaks for each of the elements. Moreover, detection limits for rare earth elements on LIF crystals as opposed to PETs tend to be significantly lower which makes this an even more complicated problem.

2) L154: Most ion microprobe studies that measure titanium contents do so with titanium mass 49. This is to avoid doubly-charged ^{96}Zr , as an isobaric interference. since the authors have chosen to use $^{48}\text{Ti}^+$, can they please comment upon whether zirconium 96 has a significant effect on their apparent titanium contents?

3) Several possibilities were explored to describe the decomposition textures noted in class 1 zircon in section 4.1. However, it is not entirely clear which one of these explanations, after reading their different possibilities, the authors prefer. It would be useful if at the conclusion of section 4.1 the authors could provide a simple short paragraph that provides clarity to their preferred interpretation for the significance of the measured dates, and put those in the context of what physical process they represent.

4) It would be useful if the authors could provide some commentary upon why they think they have identified clasts, in which the material is related in some manner of speaking to the zircon itself. Are the authors for example interpreting the clast and the zircon to be broadly the same age? Or is it also possible that the surrounding material and the zircon could have been fused together at some later time? I'm perfectly supportive of the interpretation that the authors have chosen to make; that is defining this as a clast but I would like to hear more about what makes the authors think that the zircon and the K-rich glass or silica are actually related in time of formation. The other material can't be dated, so how do the authors know it was formed contemporaneously?

5) the authors collected FTIR Spectra from 4000-650 cm^{-1} ; It would be useful to see the full spectra rather than truncating this at 1400 cm^{-1} . at the very least the authors could include the CSV raw data in the supplementary so that the interested reader can plot this up on their own.

6) With respect to table 1: why are the decomposed areas so far from 100% EPMA totals? Also the same could be asked for the non decomposed areas. Lunar zircons are fairly poor in trace elements so assuming it is related to impurities such as REEs is likely incorrect. There is likely some analytical aspect that needs further comment.

7) Figure 10 shows several examples of reversely discordant data - largely those that are referred to by the authors as 'decomposed areas.' Could the authors comment please on whether or not they believe this is in fact reversely discordant or simply apparently reversely discordant? In the case of the latter this could potentially be due to matrix effects associated with ion microprobe measurements which are presumably standardized relative to crystalline zircon, but then used to calculate discordance for something that has been disturbed. In any case one the key question

then becomes as to how common lead correction is actually done for data of this nature, as there are not really many way in which zircon can become reversely discordant.

In summary this was a nice paper and a joy to read and I look forward to seeing it in print.

Review form: Reviewer 2

Is the manuscript scientifically sound in its present form?

Yes

Are the interpretations and conclusions justified by the results?

Yes

Is the language acceptable?

Yes

Do you have any ethical concerns with this paper?

No

Have you any concerns about statistical analyses in this paper?

No

Recommendation?

Accept with minor revision (please list in comments)

Comments to the Author(s)

In lines 325 and 333, the authors use the phrase "it is challenging to interpret". recommend changing one of these for greater variation of language (especially since these are only a couple of lines apart)

In line 356: grain boundaries between k-rich phase and silica are described as "blurred" perhaps there is a more petrological term that could be used? maybe diffusional? or anhedral? diffuse? I think there must be a better word than "blurred" since that's a visual observation, but is probably not the physical condition of the grain boundaries. (but I'm away from my office working from home, and I don't have my geological dictionary with me, so I don't have the correct word to suggest).

Review form: Reviewer 3 (Marc Norman)

Is the manuscript scientifically sound in its present form?

No

Are the interpretations and conclusions justified by the results?

No

Is the language acceptable?

Yes

Do you have any ethical concerns with this paper?

No

Have you any concerns about statistical analyses in this paper?

Yes

Recommendation?

Major revision is needed (please make suggestions in comments)

Comments to the Author(s)

This is a nice study of zircons from two evolved clasts that were extracted from a lunar regolith breccia. It's a complicated and largely unresolved story, however, due to the limited data and the uncertain petrogenesis of the clasts. In my view, the primary value of this paper is in documentation of the materials as there are few new insights into broader questions of lunar crustal evolution of the impact history. Nonetheless, studies like these have their place and I would support publication pending at least one fairly major revision and a few other minor edits.

The one major revision, and the reason that I ticked NO boxes for the editorial questions "Is the manuscript scientifically sound in its present form?" and "Are the interpretations and conclusions justified by the results?" is that there appears to be a disconnect between the ages discussed in the text and that presented in Table 1. Specifically, although Line 321 states: "Our discussion about the significance of the measured dates will thus focus only on [the] 207Pb/206Pb dates", all of the dates mentioned specifically in the text are indicated as 206U/238U dates in Table 1.

For example, Line 282 states "Two of the analyses from the core of Grain 1 are concordant (0.6 to 2% discordance) and yielded 207Pb/206Pb dates of 3905 ± 14 Ma (analysis #1) and 3896 ± 17 Ma (analysis #4), and 206Pb/238U dates of 3984 ± 183 Ma (analysis #1) and 3918 ± 193 Ma (analysis #4)". However, Table 1 lists these analyses in the opposite order, i.e., Analysis 1: 6/38 = 3905 and 7/6 = 3984 Ma.

For these concordant analyses it doesn't matter a great deal but the problem becomes significant when discussing the more discordant grains, e.g., in the decomposed areas. For example, Line 294 states: "The 207Pb/206Pb apparent dates obtained on these decomposed areas are 3379 ± 37 Ma (Grain 1 analysis #5), 3592 ± 30 Ma (Grain 2 analysis #1) and 3478 ± 27 Ma (Grain 2)." However, Table 1 lists these as 6/38 ages whereas the corresponding 7/6 ages are 4.1-4.2 Ga.

I re-calculated the ages based on the isotope ratios presented in Table 1 and this appears to be a simple translation of the age data in Table 1 and the text appears to be correct. However, because this has potentially major implications for the discussion, it should be checked carefully by the authors and revised as necessary.

Another somewhat pedantic but moderately important edit is referring to the isotope data as 'ages', which implies a specifically datable event and appropriate statistics. As discussed in the text, however, some of these data may reflect partial resetting of the U-Pb isotopes and therefore have no specific geochronological significance, and in many cases the 'ages' are based on only a single point analysis rather than a population. I understand that these are the data at hand, but in no way is the in-run precision an adequate measure of the population. Greater precision in the language and a note to this effect in the text here would be appropriate.

Minor suggested edits keyed to line numbers:

77: references to the previous studies would be appropriate here, e.g., Andersen and Hinthorne (1973; 68415/6), Meyer et al. (1989 LPSC abstract), Bouvier et al. (2011 MAPS abstract; 65015), Norman and Nemchin (2014), Vanderliek et al (2018 AGU, 2017 and 2019 LPSC) and any other that the authors might know about. [note that Norman and Nemchin 2014 and Norman et al. 2016 are referenced in the text but not listed in the bibliography].

Along these same lines, the statement that few IIE-rich phases have been dated in A16 samples may be correct in fact, but it finesses the fact that many of the samples at that site are unsuitable for such studies. E.g., the granitic glasses referenced here obviously cannot be dated in this way.

Sample characteristics such as grainsize is an important limitations in this lack of data, and this could be mentioned specifically.

It would also be worth mentioning that most of the A16 samples lack such phases and by concentrating on the KREEPy samples where they are present will inevitably introduce a bias toward Imbrium in the data. Such a bias has already complicated attempts to disentangle the broader impact history of the Moon (e.g., as discussed by Norman and Nemchin 2014, Norman et al. 2016, Bottke and Norman 2017, among others), and should be recognised explicitly in studies such as this one, which also concludes that the samples record a significant Imbrium overprint.

Line 317 "This study provides the first U-Pb geochronological data for zircon from Apollo 16 samples". This broad statement is not strictly correct, see references cited above. A previous version of this statement in the Introduction "Here we report the first U-Pb dates obtained on three zircon grains found within an Apollo 16 regolith breccia sample" is more appropriate.

Line 348: Norman and Nemchin 2014 also describe partial resetting of U-Pb in zirconolite and apatite in a clast extracted from Imbrium ejecta. See also Vanderliek et al (2019 LPSC) and refs therein for similar discussion of zircon.

Line 426: "The Pb/Pb dates obtained on crystalline zircon grains (4.01 to 3.9 Ga) analysed in both clasts are consistent with these High Alkali Suite U-Pb and Pb/Pb dates and we propose that Clast 1 and the zircon in Clast 2 originate from these types of evolved magmatic parent rocks". As stated, this seems to imply that the authors consider these dates to represent the primary magmatic ages of these rocks when the text tends to suggest that they are partially to completely reset by Imbrium. As such, the correspondence of the dates with the HAS is coincidental and may not be especially significant. All this is saying is that most HAS rocks are older than Imbrium, which is mostly self-evident as many of them occur as clasts in Imbrium ejecta. The chemical and mineralogical compositions of these clasts provide a stronger link to the HAS. The petrogenetic uncertainty of these clasts is noted appropriately later in the paragraph so just a little editing of the text here to emphasise the min-pet and acknowledge the uncertainty in the U-Pb data ('e'g', although the 7/6 ages of these clasts overlap with those of the HAS, the interpretation of these ages is uncertain...etc.).

Line 440: A ref to Norman and Nemchin 2014 would be appropriate here as they specifically discuss resetting of U-Pb in zirconolite and apatite by Imbrium.

Line 470: "...a maturity indicator $Is/FeO = 27$ (Jerde et al. 1990), suggesting that it had experienced moderate amounts of time at the lunar surface (apx. timescale of tens to hundreds of millions of years of space exposure)..". This has not been discussed earlier in the text so it is not strictly a Conclusion of the study but would also suggest checking the approximate timescales of exposure as my understanding is that hundreds of millions of years is ample time to produce a mature soil. This Is is quite low and may suggest a much shorter time of exposure.

end of review.
Marc Norman

Decision letter (RSOS-200236.R0)

06-Apr-2020

Dear Dr Joy,

The editors assigned to your paper ("Timing of geological events in the lunar highlands recorded

in shocked zircon-bearing clasts from Apollo 16") have now received comments from reviewers. We would like you to revise your paper in accordance with the referee and Associate Editor suggestions which can be found below (not including confidential reports to the Editor). Please note this decision does not guarantee eventual acceptance.

Please submit a copy of your revised paper before 29-Apr-2020. Please note that the revision deadline will expire at 00.00am on this date. If we do not hear from you within this time then it will be assumed that the paper has been withdrawn. In exceptional circumstances, extensions may be possible if agreed with the Editorial Office in advance. We do not allow multiple rounds of revision so we urge you to make every effort to fully address all of the comments at this stage. If deemed necessary by the Editors, your manuscript will be sent back to one or more of the original reviewers for assessment. If the original reviewers are not available, we may invite new reviewers.

- Data accessibility

<http://datadryad.org/submit?journalID=RSOS&manu=RSOS-200236>

- Competing interests

- Authors' contributions

- Acknowledgements

- Funding statement

Many thanks & best wishes,

on behalf of the Associate Editor, and Professor Rob Ivison (Subject Editor)
openscience@royalsociety.org

Associate Editor's comments:

Thank you for this promising contribution. The view of the reviewers is that the paper has many merits, though there are a number of areas requiring revision to push the paper from being good to excellent. In particular, please carefully consider the comments of the third reviewer. We'll look forward to receiving your revision in due course.

Reviewers' Comments to Author:

Reviewer: 1

Comments to the Author(s)

This study investigated 2 Apollo 16 clasts which contain zircon. Clast 1 has a zircon interpreted to be from a high alkali suite rock, with zircon ages that range between 4.01 and 3.4 Ga. Clast 2 U-Pb ages are between 3.97 and 3.9 one Ga and is more akin to KREEP-rich impact melt breccia. Data were collected by in situ ion microprobe. Overall the authors present a nice description of the data which also includes spectroscopy measurements and trace element analysis. The

microchemical features of the grains are well described, as are the physical features including curvy planar features indicating what the authors interpret to be low pressure <20 GPa impact shocking. This is a nice contribution and will be welcome by the community and I recommend publication. With this said I do raise several considerations for the authors to address or respond to before this manuscript is published.

1) L109 The authors had provided insufficient detail to evaluate the EMPA rare earth element measurements that have been conducted on zircon. The elements that the authors have targeted by electron microprobe are extremely difficult measurements with multiple X-ray interferences and overlaps. Moreover the overlaps between the rare earth element doped glass and the zircons may not be systematic. That is, the rare earth elements doped glasses could be doped uniformly (i.e., not with relative chondrite normalized abundances) and may not therefore reflect the "odd even" effects for natural samples. While I would not normally request this for an electron microprobe measurement these measurements are so difficult that I think the authors need to provide the location of their backgrounds and the location of their peaks for each of the elements. Moreover, detection limits for rare earth elements on LIF crystals as opposed to PETs tend to be significantly lower which makes this an even more complicated problem.

2) L154: Most ion microprobe studies that measure titanium contents do so with titanium mass 49. This is to avoid doubly-charged ^{96}Zr , as an isobaric interference. since the authors have chosen to use $^{48}\text{Ti}^+$, can they please comment upon whether zirconium 96 has a significant effect on their apparent titanium contents?

3) Several possibilities were explored to describe the decomposition textures noted in class 1 zircon in section 4.1. However, it is not entirely clear which one of these explanations, after reading their different possibilities, the authors prefer. It would be useful if at the conclusion of section 4.1 the authors could provide a simple short paragraph that provides clarity to their preferred interpretation for the significance of the measured dates, and put those in the context of what physical process they represent.

4) It would be useful if the authors could provide some commentary upon why they think they have identified clasts, in which the material is related in some manner of speaking to the zircon itself. Are the authors for example interpreting the clast and the zircon to be broadly the same age? Or is it also possible that the surrounding material and the zircon could have been fused together at some later time? I'm perfectly supportive of the interpretation that the authors have chosen to make; that is defining this as a clast but I would like to hear more about what makes the authors think that the zircon and the K-rich glass or silica are actually related in time of formation. The other material can't be dated, so how do the authors know it was formed contemporaneously?

5) the authors collected FTIR Spectra from 4000-650 cm^{-1} ; It would be useful to see the full spectra rather than truncating this at 1400 cm^{-1} . at the very least the authors could include the CSV raw data in the supplementary so that the interested reader can plot this up on their own.

6) With respect to table 1: why are the decomposed areas so far from 100% EPMA totals? Also the same could be asked for the non decomposed areas. Lunar zircons are fairly poor in trace elements so assuming it is related to impurities such as REEs is likely incorrect. There is likely some analytical aspect that needs further comment.

7) Figure 10 shows several examples of reversely discordant data - largely those that are referred to by the authors as 'decomposed areas.' Could the authors comment please on whether or not they believe this is in fact reversely discordant or simply apparently reversely discordant? In the case of the latter this could potentially be due to matrix effects associated with ion microprobe measurements which are presumably standardized relative to crystalline zircon, but then used to calculate discordance for something that has been disturbed. In any case one the key question then becomes as to how common lead correction is actually done for data of this nature, as there are not really many way in which zircon can become reversely discordant.

In summary this was a nice paper and a joy to read and I look forward to seeing it in print.

Reviewer: 2

Comments to the Author(s)

In lines 325 and 333, the authors use the phrase "it is challenging to interpret". recommend changing one of these for greater variation of language (especially since these are only a couple of lines apart)

In line 356: grain boundaries between k-rich phase and silica are described as "blurred" perhaps there is a more petrological term that could be used? maybe diffusional? or anhedral? diffuse? I think there must be a better word than "blurred" since that's a visual observation, but is probably not the physical condition of the grain boundaries. (but I'm away from my office working from home, and I don't have my geological dictionary with me, so I don't have the correct word to suggest).

Reviewer: 3

Comments to the Author(s)

This is a nice study of zircons from two evolved clasts that were extracted from a lunar regolith breccia. It's a complicated and largely unresolved story, however, due to the limited data and the uncertain petrogenesis of the clasts. In my view, the primary value of this paper is in documentation of the materials as there are few new insights into broader questions of lunar crustal evolution of the impact history. Nonetheless, studies like these have their place and I would support publication pending at least one fairly major revision and a few other minor edits.

The one major revision, and the reason that I ticked NO boxes for the editorial questions "Is the manuscript scientifically sound in its present form?" and "Are the interpretations and conclusions justified by the results?" is that there appears to be a disconnect between the ages discussed in the text and that presented in Table 1. Specifically, although Line 321 states: "Our discussion about the significance of the measured dates will thus focus only on [the] 207Pb/206Pb dates", all of the dates mentioned specifically in the text are indicated as 206U/238U dates in Table 1.

For example, Line 282 states "Two of the analyses from the core of Grain 1 are concordant (0.6 to 2% discordance) and yielded 207Pb/206Pb dates of 3905 ± 14 Ma (analysis #1) and 3896 ± 17 Ma (analysis #4), and 206Pb/238U dates of 3984 ± 183 Ma (analysis #1) and 3918 ± 193 Ma (analysis #4)". However, Table 1 lists these analyses in the opposite order, i.e., Analysis 1: 6/38 = 3905 and 7/6 = 3984 Ma.

For these concordant analyses it doesn't matter a great deal but the problem becomes significant when discussing the more discordant grains, e.g., in the decomposed areas. For example, Line 294 states: "The 207Pb/206Pb apparent dates obtained on these decomposed areas are 3379 ± 37 Ma (Grain 1 analysis #5), 3592 ± 30 Ma (Grain 2 analysis #1) and 3478 ± 27 Ma (Grain 2)." However, Table 1 lists these as 6/38 ages whereas the corresponding 7/6 ages are 4.1-4.2 Ga.

I re-calculated the ages based on the isotope ratios presented in Table 1 and this appears to be a simple translation of the age data in Table 1 and the text appears to be correct. However, because this has potentially major implications for the discussion, it should be checked carefully by the authors and revised as necessary.

Another somewhat pedantic but moderately important edit is referring to the isotope data as 'ages', which implies a specifically datable event and appropriate statistics. As discussed in the text, however, some of these data may reflect partial resetting of the U-Pb isotopes and therefore

have no specific geochronological significance, and in many cases the 'ages' are based on only a single point analysis rather than a population. I understand that these are the data at hand, but in no way is the in-run precision an adequate measure of the population. Greater precision in the language and a note to this effect in the text here would be appropriate.

Minor suggested edits keyed to line numbers:

77: references to the previous studies would be appropriate here, e.g., Andersen and Hinthorne (1973; 68415/6), Meyer et al. (1989 LPSC abstract), Bouvier et al. (2011 MAPS abstract; 65015), Norman and Nemchin (2014), Vanderliek et al (2018 AGU, 2017 and 2019 LPSC) and any other that the authors might know about. [note that Norman and Nemchin 2014 and Norman et al. 2016 are referenced in the text but not listed in the bibliography].

Along these same lines, the statement that few ITE-rich phases have been dated in A16 samples may be correct in fact, but it finesses the fact that many of the samples at that site are unsuitable for such studies. E.g., the granitic glasses referenced here obviously cannot be dated in this way. Sample characteristics such as grainsize is an important limitations in this lack of data, and this could be mentioned specifically.

It would also be worth mentioning that most of the A16 samples lack such phases and by concentrating on the KREEPy samples where they are present will inevitably introduce a bias toward Imbrium in the data. Such a bias has already complicated attempts to disentangle the broader impact history of the Moon (e.g., as discussed by Norman and Nemchin 2014, Norman et al. 2016, Bottke and Norman 2017, among others), and should be recognised explicitly in studies such as this one, which also concludes that the samples record a significant Imbrium overprint.

Line 317 "This study provides the first U-Pb geochronological data for zircon from Apollo 16 samples". This broad statement is not strictly correct, see references cited above. A previous version of this statement in the Introduction "Here we report the first U-Pb dates obtained on three zircon grains found within an Apollo 16 regolith breccia sample" is more appropriate.

Line 348: Norman and Nemchin 2014 also describe partial resetting of U-Pb in zirconolite and apatite in a clast extracted from Imbrium ejecta. See also Vanderliek et al (2019 LPSC) and refs therein for similar discussion of zircon.

Line 426: "The Pb/Pb dates obtained on crystalline zircon grains (4.01 to 3.9 Ga) analysed in both clasts are consistent with these High Alkali Suite U-Pb and Pb/Pb dates and we propose that Clast 1 and the zircon in Clast 2 originate from these types of evolved magmatic parent rocks". As stated, this seems to imply that the authors consider these dates to represent the primary magmatic ages of these rocks when the text tends to suggest that they are partially to completely reset by Imbrium. As such, the correspondence of the dates with the HAS is coincidental and may not be especially significant. All this is saying is that most HAS rocks are older than Imbrium, which is mostly self-evident as many of them occur as clasts in Imbrium ejecta. The chemical and mineralogical compositions of these clasts provide a stronger link to the HAS. The petrogenetic uncertainty of these clasts is noted appropriately later in the paragraph so just a little editing of the text here to emphasise the min-pet and acknowledge the uncertainty in the U-Pb data ('e.g.', although the 7/6 ages of these clasts overlap with those of the HAS, the interpretation of these ages is uncertain...etc.).

Line 440: A ref to Norman and Nemchin 2014 would be appropriate here as they specifically discuss resetting of U-Pb in zirconolite and apatite by Imbrium.

Line 470: "...a maturity indicator $Is/FeO = 27$ (Jerde et al. 1990), suggesting that it had experienced moderate amounts of time at the lunar surface (apx. timescale of tens to hundreds of millions of years of space exposure)." This has not been discussed earlier in the text so it is not strictly a Conclusion of the study but would also suggest checking the approximate timescales of exposure

as my understanding is that hundreds of millions of years is ample time to produce a mature soil. This is quite low and may suggest a much shorter time of exposure.

end of review.

Marc Norman

Author's Response to Decision Letter for (RSOS-200236.R0)

See Appendix A.

RSOS-200236.R1 (Revision)

Review form: Reviewer 3 (Marc Norman)

Is the manuscript scientifically sound in its present form?

Yes

Are the interpretations and conclusions justified by the results?

Yes

Is the language acceptable?

Yes

Do you have any ethical concerns with this paper?

No

Have you any concerns about statistical analyses in this paper?

No

Recommendation?

Accept as is

Comments to the Author(s)

I thank the authors for their constructive revisions to the manuscript. In my opinion, they have adequately addressed all of the outstanding issues raised in the first round of reviews and am pleased to recommend acceptance of the paper in its current form.

In preparation of the final version, the authors might wish to consider a few minor points:

1. The response does not really address the concerns of Reviewer 1 at Line 109. The concern of the reviewer was mainly in the potential for multiple interferences on the REE analyses by EMP. This is a well known and long-standing technical problem with analyses of REE by EMP, and one reason that Mike Drake and others developed a set of REE-specific glasses for calibration of these elements for analysis by EMP. The current reply only addresses detection limits and provides the EMP methods tables in the supplementary file (which are not all that useful) but it does not address concerns over the interferences directly. While it is true that most of the REE data are below detection limits, to some extent the authors are making it more difficult on themselves than necessary because the potential interferences influence the DL's, but these are obviously difficult

to de-convolve. An alternative approach would be to simply report the data that they were able to obtain. I understand the potential utility of reporting DL's but in this case the DL's for REE in zircon by EMP analysis are not that useful and could be deleted with no loss of useful information.

2. The reply to Reviewer 1's concern about the poor totals of the EMP analyses of the decomposed areas (point #6) refers to the caption of Fig. 1. However, Fig. 1 is a global map of Th on the nearside and does not refer to the EMP data. I cannot find any mention of poor totals in any figure caption, and in any case this material would be better suited to the main text rather than a figure caption. This needs to be addressed directly and specifically.

3. It would be useful to include a stronger statement of the reply to point #7 from Reviewer 1 regarding the discordant U-Pb data in the main text, including the caveat regarding definition of the lunar common Pb. I see a passing statement about this but additional commentary in either the main text or the figure captions would be appropriate.

4. I would caution against using the term 'devitrified glass'. The term has a long and storied history in lunar sample petrography, but in detail it is a poor description because it implies initial formation of vitric glass and then a subsequent process of devitrification. On Earth, devitrification is due to atomic-scale dissolution of the glass by water and subsequent precipitation, such that actual glass rarely survives for more than a few Ma, if that. This is unlikely to be a significant process on the Moon where genuine glass apparently persists in the regolith for billions of years. So unless the authors have specific evidence for devitrification (probably by thermal events?), I would suggest using another term such as mesostasis, fine-grained quench textures, etc.

I look forward to seeing the final version of the paper.

Best regards,

Marc Norman, RSES-ANU

Decision letter (RSOS-200236.R1)

Dear Dr Joy,

It is a pleasure to accept your manuscript entitled "Timing of geological events in the lunar highlands recorded in shocked zircon-bearing clasts from Apollo 16" in its current form for publication in Royal Society Open Science. The comments of the reviewer(s) who reviewed your manuscript are included at the foot of this letter.

on behalf of Prof Rob Ivison (Subject Editor)
openscience@royalsociety.org

Associate Editor Comments to Author:

As you'll see, the reviewer is largely satisfied by the changes you have made in response to their comments, and those of the other reviewers; however, they recommend a few remaining modifications to elevate the manuscript further.

Reviewer comments to Author:

Reviewer: 3

Comments to the Author(s)

I thank the authors for their constructive revisions to the manuscript. In my opinion, they have adequately addressed all of the outstanding issues raised in the first round of reviews and am pleased to recommend acceptance of the paper in its current form.

In preparation of the final version, the authors might wish to consider a few minor points:

1. The response does not really address the concerns of Reviewer 1 at Line 109. The concern of the reviewer was mainly in the potential for multiple interferences on the REE analyses by EMP. This is a well known and long-standing technical problem with analyses of REE by EMP, and one reason that Mike Drake and others developed a set of REE-specific glasses for calibration of these elements for analysis by EMP. The current reply only addresses detection limits and provides the EMP methods tables in the supplementary file (which are not all that useful) but it does not address concerns over the interferences directly. While it is true that most of the REE data are below detection limits, to some extent the authors are making it more difficult on themselves than necessary because the potential interferences influence the DL's, but these are obviously difficult to de-convolve. An alternative approach would be to simply report the data that they were able to obtain. I understand the potential utility of reporting DL's but in this case the DL's for REE in zircon by EMP analysis are not that useful and could be deleted with no loss of useful information.
2. The reply to Reviewer 1's concern about the poor totals of the EMP analyses of the decomposed areas (point #6) refers to the caption of Fig. 1. However, Fig. 1 is a global map of Th on the nearside and does not refer to the EMP data. I cannot find any mention of poor totals in any figure caption, and in any case this material would be better suited to the main text rather than a figure caption. This needs to be addressed directly and specifically.
3. It would be useful to include a stronger statement of the reply to point #7 from Reviewer 1 regarding the discordant U-Pb data in the main text, including the caveat regarding definition of

the lunar common Pb. I see a passing statement about this but additional commentary in either the main text or the figure captions would be appropriate.

4. I would caution against using the term 'devitrified glass'. The term has a long and storied history in lunar sample petrography, but in detail it is a poor description because it implies initial formation of vitric glass and then a subsequent process of devitrification. On Earth, devitrification is due to atomic-scale dissolution of the glass by water and subsequent precipitation, such that actual glass rarely survives for more than a few Ma, if that. This is unlikely to be a significant process on the Moon where genuine glass apparently persists in the regolith for billions of years. So unless the authors have specific evidence for devitrification (probably by thermal events?), I would suggest using another term such as mesostasis, fine-grained quench textures, etc.

I look forward to seeing the final version of the paper.

Best regards,

Marc Norman, RSES-ANU

Appendix A

Department of Earth and Environmental Sciences
University of Manchester
Williamson Building, Oxford Road
Manchester, M13 9PL, UK

28th April 2020

Dear Professor Rob Ivison,

Astronomy Subject Editor *Royal Society Open Science*

Thank you for handling our manuscript and organising the review process.

Please find attached a revised manuscript for the *Royal Society Open Science New Talent Astronomy special collection*. The original research article entitled “Timing of geological events in the lunar highlands recorded in shocked zircon-bearing clasts from Apollo 16” is within the field of Solar System science, and investigates the effects of impact bombardment on rocks collected from the Moon. We have tracked the changes to show what has been modified from the original submitted version to take into account the reviewer’s comments. The manuscript is accompanied by a Supplementary text file (with figures) and a Supplementary Excel data file.

Please also find the responses to the reviewer’s constructive comments below.

If you have any questions about our submission, please don’t hesitate to contact me.

Yours sincerely,

Dr Katherine Joy

Associate Editor's comments:

Thank you for this promising contribution. The view of the reviewers is that the paper has many merits, though there are a number of areas requiring revision to push the paper from being good to excellent. In particular, please carefully consider the comments of the third reviewer. We'll look forward to receiving your revision in due course.

Please find our responses to the reviewer's comments in red font. In addition to addressing the issues raised by the reviewers, we have also added a few additional relevant references, and acknowledge the reviewers in helping us to develop a better manuscript.

Reviewers' Comments to Author:

Reviewer: 1

Comments to the Author(s)

This study investigated 2 Apollo 16 clasts which contain zircon. Clast 1 has a zircon interpreted to be from a high alkali suite rock, with zircon ages that range between 4.01 and 3.4 Ga. Clast 2 U-Pb ages are between 3.97 and 3.9 one Ga and is more akin to KREEP-rich impact melt breccia. Data were collected by in situ ion microprobe. Overall the authors present a nice description of the data which also includes spectroscopy measurements and trace element analysis. The microchemical features of the grains are well described, as are the physical features including curvy planar features indicating what the authors interpret to be low pressure <20 GPa impact shocking. This is a nice contribution and will be welcome by the community and I recommend publication.

We thank the reviewer for their positive comments.

With this said I do raise several considerations for the authors to address or respond to before this manuscript is published.

1) L109 The authors had provided insufficient detail to evaluate the EMPA rare earth element measurements that have been conducted on zircon. The elements that the authors have targeted by electron microprobe are extremely difficult measurements with multiple X-ray interferences and overlaps. Moreover the overlaps between the rare earth element doped glass and the zircons may not be systematic. That is, the rare earth elements doped glasses could be doped uniformly (i.e., not with relative chondrite normalized abundances) and may not therefore reflect the "odd even" effects for natural samples. While I would not normally request this for an electron microprobe measurement these measurements are so difficult that I think the authors need to provide the location of their backgrounds and the location of their peaks for each of the elements. Moreover, detection limits for rare earth elements on LIF crystals as opposed to PETs tend to be significantly lower which makes this an even more complicated problem.

As stated in the methods section only data above the detection limits are reported as part of our standard data reduction process. We measured more elements than we report in the paper during the analytical session, but many, as the reviewer points out have detection limits higher than the value reported. As a consequence, all the elements measured on the LLIF crystal (Pb, U, Nd, Eu, Sm and La) were below the detection limits and were not reported in Table S3.

We updated Tables 1 and S3 to include some previously non reported ThO₂ data. In Table S3 we now report the typical errors for each element in oxide wt%., and to fix a minor error in the HfO₂ cation calculation.

Full details of the EPMA setup has added to the supplementary section of the paper as a note (though this is more details than we are usually asked to provide to support the EMPA methodology).

2) L154: Most ion microprobe studies that measure titanium contents do so with titanium mass 49. This is to avoid doubly-charged ^{96}Zr , as an isobaric interference. since the authors have chosen to use $^{48}\text{Ti}^+$, can they please comment upon whether zirconium 96 has a significant effect on their apparent titanium contents?

As shown in Figure 7 by comparing the $^{48}\text{Ti}^+$ and $^{94}\text{Zr}^+$ NanoSIMS maps, it is clear that interference from doubly charged ^{96}Zr did not affect ^{48}Ti analysis. This is because (i) analyses were carried out with a fairly high mass resolving power, (ii) ^{96}Zr only accounts for 2.8% of Zr, and (iii) unlike ICP-MS, doubly charged species are not typically detected on the NanoSIMS.

3) Several possibilities were explored to describe the decomposition textures noted in class 1 zircon in section 4.1. However, it is not entirely clear which one of these explanations, after reading their different possibilities, the authors prefer. It would be useful if at the conclusion of section 4.1 the authors could provide a simple short paragraph that provides clarity to their preferred interpretation for the significance of the measured dates, and put those in the context of what physical process they represent.

A short paragraph has been added at the end of section 4.1 and see the discussion in section 5 (disturbance at 3.4 Ga section).

4) It would be useful if the authors could provide some commentary upon why they think they have identified clasts, in which the material is related in some manner of speaking to the zircon itself. Are the authors for example interpreting the clast and the zircon to be broadly the same age? Or is it also possible that the surrounding material and the zircon could have been fused together at some later time? I'm perfectly supportive of the interpretation that the authors have chosen to make; that is defining this as a clast but I would like to hear more about what makes the authors think that the zircon and the K-rich glass or silica are actually related in time of formation. The other material can't be dated, so how do the authors know it was formed contemporaneously?

We have added a sentence at the start of 4.1 to discuss the relationships in clast 1, and a sentence 4.2 to discuss this for clast 2.

5) the authors collected FTIR Spectra from 4000-650 cm^{-1} ; It would be useful to see the full spectra rather than truncating this at 1400 cm^{-1} . at the very least the authors could include the CSV raw data in the supplementary so that the interested reader can plot this up on their own.

The full FTIR spectral data were provided in full in the Electronic Supplement Table S4, and were already available to the reviewers as part of the original paper submissions. The figures show the relevant region of interest needed for the discussion, and as such we believe that there is no need to plot the full spectral range.

6) With respect to table 1: why are the decomposed areas so far from 100% EPMA totals? Also the same could be asked for the non decomposed areas. Lunar zircons are fairly poor in trace elements so assuming it is related to impurities such as REEs is likely incorrect. There is likely some analytical aspect that needs further comment.

It could be that the decomposed areas contain a hydrous component that was not measured, that we were missing an element in the analytical setup, or given the very fine grain size of the polymineralic phase, the analysis may be effected by unequal host phase density effects (i.e., there is a mis-match to the pure mineral standards used). We do make this point already in the figure caption of Figure 1.

7) Figure 10 shows several examples of reversely discordant data - largely those that are referred to by the authors as 'decomposed areas.' Could the authors comment please on whether or not they believe this is in fact reversely discordant or simply apparently reversely discordant? In the case of the latter this could potentially be due to matrix effects associated with ion microprobe measurements which are presumably standardized relative to crystalline zircon, but then used to calculate discordance for something that has been disturbed. In any case one of the key questions then becomes as to how common lead correction is actually done for data of this nature, as there are not really many ways in which zircon can become reversely discordant.

We consider the reverse discordance to be simply an artefact of the grain alteration away from crystalline zircon and the reviewer is indeed correct that true reverse discordance is not really likely. However, this affects only the elemental (i.e. U/Pb) ratios and derived ages; within analytical uncertainty, Pb isotope data are unaffected by such matrix issues in SIMS and so the calculation of a common Pb correction, which only uses Pb isotope ratios, remains valid, subject to the usual caveat of estimating reliably what the common Pb composition should be.

In summary this was a nice paper and a joy to read and I look forward to seeing it in print.

Thank you.

Reviewer: 2

Comments to the Author(s)

In lines 325 and 333, the authors use the phrase "it is challenging to interpret". recommend changing one of these for greater variation of language (especially since these are only a couple of lines apart)

The phrasing of the second use of this language has been modified.

In line 356: grain boundaries between k-rich phase and silica are described as "blurred" perhaps there is a more petrological term that could be used? maybe diffusional? or anhedral? diffuse? I think there must be a better word than "blurred" since that's a visual observation, but is probably not the physical condition of the grain boundaries. (but I'm away from my office working from home, and I don't have my geological dictionary with me, so I don't have the correct word to suggest).

The wording has been modified.

Reviewer: 3

Comments to the Author(s)

This is a nice study of zircons from two evolved clasts that were extracted from a lunar regolith breccia. It's a complicated and largely unresolved story, however, due to the limited data and the uncertain petrogenesis of the clasts. In my view, the primary value of this paper is in documentation of the materials as there are few new insights into broader questions of lunar crustal evolution of the impact history. Nonetheless, studies like these have their place and I would support publication pending at least one fairly major revision and a few other minor edits.

The one major revision, and the reason that I ticked NO boxes for the editorial questions "Is the manuscript scientifically sound in its present form?" and "Are the interpretations and conclusions justified by the results?" is that there appears to be a disconnect between the ages discussed in the text and that presented in Table 1. Specifically, although Line 321 states: "Our discussion about the

significance of the measured dates will thus focus only on [the] $^{207}\text{Pb}/^{206}\text{Pb}$ dates", all of the dates mentioned specifically in the text are indicated as $^{206}\text{U}/^{238}\text{U}$ dates in Table 1.

For example, Line 282 states "Two of the analyses from the core of Grain 1 are concordant (0.6 to 2% discordance) and yielded $^{207}\text{Pb}/^{206}\text{Pb}$ dates of 3905 ± 14 Ma (analysis #1) and 3896 ± 17 Ma (analysis #4), and $^{206}\text{Pb}/^{238}\text{U}$ dates of 3984 ± 183 Ma (analysis #1) and 3918 ± 193 Ma (analysis #4)". However, Table 1 lists these analyses in the opposite order, i.e., Analysis 1: $6/38 = 3905$ and $7/6 = 3984$ Ma. For these concordant analyses it doesn't matter a great deal but the problem becomes significant when discussing the more discordant grains, e.g., in the decomposed areas. For example, Line 294 states: "The $^{207}\text{Pb}/^{206}\text{Pb}$ apparent dates obtained on these decomposed areas are 3379 ± 37 Ma (Grain 1 analysis #5), 3592 ± 30 Ma (Grain 2 analysis #1) and 3478 ± 27 Ma (Grain 2)." However, Table 1 lists these as $6/38$ ages whereas the corresponding $7/6$ ages are 4.1-4.2 Ga. I recalculated the ages based on the isotope ratios presented in Table 1 and this appears to be a simple translation of the age data in Table 1 and the text appears to be correct. However, because this has potentially major implications for the discussion, it should be checked carefully by the authors and revised as necessary.

Good spot – many thanks for the attention to detail by Marc. We have amended the text as the analyses in the manuscript wording was correct, but the table column labels had been accidentally switched when the table was being reformatted for final submission. Sorry about this confusion. The words in the discussion and manuscript do not need to be amended.

In correcting this error we also realised that the reported $^{207}/^{206}$ ages of Clast 2, Spot 2 and Clast 2, Spot 3 were copied over from our original data table incorrectly into the formatted Table 2. The numbers cited in the text and displayed on all the figures were (and still are correct), however, in the Table 2 itself were not right (they were a duplication of the Clast 1, Spot 1 and Clast, Spot 2 analyses). We have corrected this error as well.

Another somewhat pedantic but moderately important edit is referring to the isotope data as 'ages', which implies a specifically datable event and appropriate statistics. As discussed in the text, however, some of these data may reflect partial resetting of the U-Pb isotopes and therefore have no specific geochronological significance, and in many cases the 'ages' are based on only a single point analysis rather than a population. I understand that these are the data at hand, but in no way is the in-run precision an adequate measure of the population. Greater precision in the language and a note to this effect in the text here would be appropriate.

We agree with reviewer 3 here, and thought we originally had taken care of avoiding confusion by using 'age' in interpreting numbers to specific events (e.g., crystallisation or resetting age), and 'date' when referring to numerical values without any interpretation associated with these values. We found only a couple of instances in the revised ms where this required modification, and have made the changes.

Minor suggested edits keyed to line numbers:

77: references to the previous studies would be appropriate here, e.g., Andersen and Hinthorne (1973; 68415/6), Meyer et al. (1989 LPSC abstract), Bouvier et al. (2011 MAPS abstract; 65015), Norman and Nemchin (2014), Vanderliek et al (2018 AGU, 2017 and 2019 LPSC) and any other that the authors might know about. [note that Norman and Nemchin 2014 and Norman et al. 2016 are referenced in the text but not listed in the bibliography].

We added some additional references, however, Bouvier discusses the ages of a bulk rock sample not specifically zircon and has not been added here, but we do include this reference later on. We updated the reference list for the missing references – thanks for spotting this.

Along these same lines, the statement that few ITE-rich phases have been dated in A16 samples may be correct in fact, but it finesses the fact that many of the samples at that site are unsuitable for such studies. E.g., the granitic glasses referenced here obviously cannot be dated in this way. Sample characteristics such as grainsize is an important limitations in this lack of data, and this could be mentioned specifically.

Point noted, and some words added to the manuscript. In theory it might be possible to date the granite glass and K-Feldspar using the Stockholm group's Pb-isotope method (Snape et al. approach), but the phases look a bit too small and contaminated with sulphur blebs to get a clean SIMS spot.

It would also be worth mentioning that most of the A16 samples lack such phases and by concentrating on the KREEPy samples where they are present will inevitably introduce a bias toward Imbrium in the data. Such a bias has already complicated attempts to disentangle the broader impact history of the Moon (e.g., as discussed by Norman and Nemchin 2014, Norman et al. 2016, Bottke and Norman 2017, among others), and should be recognised explicitly in studies such as this one, which also concludes that the samples record a significant Imbrium overprint.

Sentence revised to now address this point. The Imbrium overprint issue is discussed in the final section.

Line 317 "This study provides the first U-Pb geochronological data for zircon from Apollo 16 samples". This broad statement is not strictly correct, see references cited above. A previous version of this statement in the Introduction "Here we report the first U-Pb dates obtained on three zircon grains found within an Apollo 16 regolith breccia sample" is more appropriate.

Sentence revised.

Line 348: Norman and Nemchin 2014 also describe partial resetting of U-Pb in zirconolite and apatite in a clast extracted from Imbrium ejecta. See also Vanderliek et al (2019 LPSC) and refs therein for similar discussion of zircon.

Sentence revised. We note though that none of the Vanderliek studies is not actually peer reviewed / published yet.

Line 426: "The Pb/Pb dates obtained on crystalline zircon grains (4.01 to 3.9 Ga) analysed in both clasts are consistent with these High Alkali Suite U-Pb and Pb/Pb dates and we propose that Clast 1 and the zircon in Clast 2 originate from these types of evolved magmatic parent rocks". As stated, this seems to imply that the authors consider these dates to represent the primary magmatic ages of these rocks when the text tends to suggest that they are partially to completely reset by Imbrium. As such, the correspondence of the dates with the HAS is coincidental and may not be especially significant. All this is saying is that most HAS rocks are older than Imbrium, which is mostly self-evident as many of them occur as clasts in Imbrium ejecta. The chemical and mineralogical compositions of these clasts provide a stronger link to the HAS. The petrogenetic uncertainty of these clasts is noted appropriately later in the paragraph so just a little editing of the text here to emphasise the min-pet and acknowledge the uncertainty in the U-Pb data ('e.g.', although the 7/6 ages of these clasts overlap with those of the HAS, the interpretation of these ages is uncertain...etc.).

We agree and have rephrased this paragraph to address this point.

Line 440: A ref to Norman and Nemchin 2014 would be appropriate here as they specifically discuss resetting of U-Pb in zirconolite and apatite by Imbrium.

Added.

Line 470: "...a maturity indicator $Is/FeO = 27$ (Jerde et al. 1990), suggesting that it had experienced moderate amounts of time at the lunar surface (apx. timescale of tens to hundreds of millions of years of space exposure)..". This has not been discussed earlier in the text so it is not strictly a Conclusion of the study but would also suggest checking the approximate timescales of exposure as my understanding is that hundreds of millions of years is ample time to produce a mature soil. This Is is quite low and may suggest a much shorter time of exposure.

We agree. Sentence changed to be tens of millions of years.

end of review.

Marc Norman